# Sequential vaccinations with divergent H1N1 influenza virus strains induce multi-H1 clade neutralizing antibodies in swine

Kristien Van Reeth [1] ✉, Anna Parys[1], José Carlos Mancera Gracia[1], Ivan Trus [2], Koen Chiers [3], Philip Meade[4,5], Sean Liu [4,6], Peter Palese [4], Florian Krammer [4,5,7] & Elien Vandoorn[1]

Vaccines that protect against any H1N1 influenza A virus strain would be advantageous for use in pigs and humans. Here, we try to induce a pan-H1N1 antibody response in pigs by sequential vaccination with antigenically divergent H1N1 strains. Adjuvanted whole inactivated vaccines are given intramuscularly in various two- and three-dose regimens. Three doses of heterologous monovalent H1N1 vaccine result in seroprotective neutralizing antibodies against 71% of a diverse panel of human and swine H1 strains, detectable antibodies against 88% of strains, and sterile cross-clade immunity against two heterologous challenge strains. This strategy outperforms any two-dose regimen and is as good or better than giving three doses of matched trivalent vaccine. Neutralizing antibodies are H1-specific, and the second heterologous booster enhances reactivity with conserved epitopes in the HA head. We show that even the most traditional influenza vaccines can offer surprisingly broad protection if they are administered in an alternative way.

Seasonal influenza vaccines are designed to protect against H1N1 and H3N2 influenza A virus (IAV) strains of which the major surface protein, hemagglutinin (HA), matches with that of the vaccine strains. HA is composed of a globular head that contains the receptor-binding site, which allows the virus to attach to receptors on host cells, and a stalk, which is anchored in the viral membrane. Neuraminidase (NA), the second surface protein, is a receptor-destroying enzyme that facilitates the release of newly synthesized virus particles from infected cells. While antibodies against NA and internal viral proteins may contribute to protection against infection, only antibodies against the HA head can neutralize extracellular virus and prevent viral entry into cells[1,2]. These antibodies are routinely measured by hemagglutination-inhibition (HI) assays, and they bind to five distinct antigenic sites in the HA head[3]. These antigenic sites are immunodominant over other antigenic regions in the HA head and the stalk, and the hierarchy between them is dynamic and little understood[4–6]. Because the antigenic sites are hypervariable, the IAV strains in seasonal influenza vaccines need to be reconsidered every year, to avoid antigenic mismatches with the circulating strains. These vaccines also fail to protect against pandemic and most zoonotic IAVs. Interestingly, the HA stalk is one of the most attractive targets for universal influenza vaccines, as it is conserved within a given HA subtype and between subtypes of the same phylogenetic HA group[7]. Unfortunately, stalk-specific antibodies are more difficult to induce, and they mostly affect the virus after its entry into the host cell endosome[2].

Sequential vaccination with antigenically distinct strains of the same IAV subtype may favor a secondary immune response against epitopes that are conserved between the heterologous strains and

[1]Laboratory of Virology, Faculty of Veterinary Medicine, Ghent University, Gent, Belgium. [2]Dioscuri Centre for RNA-Protein Interactions in Human Health and Disease, International Institute of Molecular and Cell Biology, Warsaw, Poland. [3]Laboratory of Pathology, Faculty of Veterinary Medicine, Ghent University, Gent, Belgium. [4]Department of Microbiology, Icahn School of Medicine at Mount Sinai, New York, NY, USA. [5]Center for Vaccine Research and Pandemic Preparedness (C-VARPP), Icahn School of Medicine at Mount Sinai, New York, NY, USA. [6]Division of Infectious Diseases, Icahn School of Medicine at Mount Sinai, New York, NY, USA. [7]Department of Pathology, Molecular and Cell Based Medicine, Icahn School of Medicine at Mount Sinai, New York, NY, USA. ✉e-mail: Kristien.VanReeth@UGent.be

increase the breadth of the anti-HA antibody repertoire[8]. The 2009 H1N1 pandemic can be seen as one of the largest natural heterologous prime-boost experiments. The HA of the 2009 H1N1 pandemic virus (H1N1pdm09) was remarkably divergent from that of human seasonal H1N1 viruses from 1977 to 2008 and most closely related to the 1918 H1N1 pandemic virus, which had been preserved in swine populations in North America[9–11]. Thus, people born after 1977 initially had little or no cross-reactive antibodies, but they rapidly developed broad, pan-H1 neutralizing antibody responses upon infection or vaccination with H1N1pdm09, which may have contributed to the extinction of earlier seasonal H1N1 viruses[8,12–14]. Antibodies to conserved epitopes in the HA stalk and head have been reported in some individuals, but their immunization histories remain uncertain because humans encounter numerous IAV strains and get multiple vaccinations during their lifetime.

The same IAV subtypes as those circulating in humans—H1N1 and H3N2—are enzootic in swine, and inactivated influenza vaccines are most widely used in both species[15]. In addition, there is bidirectional virus transmission between pigs and humans, and the surface proteins of most swine IAVs are derived from viruses that once circulated in humans[16,17]. Swine IAVs also pose a risk for reintroduction into the human population once immunity has waned sufficiently, as occurred in 2009. As many as three evolutionary distinct H1 lineages and multiple clades within each lineage are circulating simultaneously in swine, and a total of 18 H1 clades were reported in 2016–2019[18,19]. The HAs of H1 swine IAV lineages originate from the 1918 or 2009 pandemic H1N1 viruses (1A or classical swine lineage), from human seasonal H1N1 variants from 1983 to 2003 (1B or human seasonal swine lineage) and from H1N1 viruses from wild birds (1C or Eurasian avian lineage). Each H1 can be paired with diverse N1 or N2 NAs, and the prevailing lineages and clades differ between continents and regions. Because of this extraordinary diversity, no single H1 vaccine strain can induce sufficiently broad immunity and a multivalent swine influenza vaccine with all circulating variants is not feasible.

We use the pig as a model for research on more broadly protective vaccines and vaccination strategies for humans as well as swine[20]. Here, we try to induce a pan-H1N1 antibody response by prime-boost immunization with whole inactivated, adjuvanted vaccines based on distinct H1 swine IAV lineages. Apart from antibodies against the HA head, we assess antibodies against the HA stalk, the NA and known H1 antigenic sites. Unlike two administrations of heterologous monovalent vaccine, three sequential administrations induce detectable neutralizing antibodies against 88% of a diverse panel of swine and human H1 virus strains. This strategy outperforms any two-dose regimen and three doses of homologous monovalent or trivalent vaccine. The multi-clade neutralizing antibodies are mainly directed against known, hypervariable antigenic sites in the HA head, but three-dose heterologous prime-boost vaccination tends to enhance reactivity with conserved epitopes in the HA head. Our results suggest that changes in vaccine regimens and administration policies could tremendously enhance the potency of existing influenza vaccines in both swine and humans.

## Results

### Antibodies against vaccine strains after two-dose vaccination

We prepared adjuvanted inactivated, monovalent whole virus vaccines based on four antigenically distinct H1N1 IAV strains: the prototype H1N1pdm09 virus strain A/California/04/2009 (CA09, clade 1A.3.3.2), and swine IAV strains of the European human-like (A/swine/Cotes d'Armor/0046/2008, ARM08, clade 1B.1.2.3), the North American human-like (A/swine/Illinois/00685/2005, IL05, clade 1B.2.1), and the Eurasian avian (A/swine/Gent/28/2010, G10, clade 1C.2.1) lineage. CA09, ARM08 and G10 represent the prevailing H1 clades in swine in Europe, and descendants of CA09 are circulating in swine and humans worldwide. The vaccine strains show extensive genetic and antigenic

diversity in the HA1 subunit of the HA and less diversity in the NA (Fig. 1, Supplementary Tables 1–7). In a first experiment, pigs were vaccinated with one of the four vaccine strains and boosted with a different strain 4 weeks later. We tested eight of the 12 possible heterologous prime-boost combinations, including two combinations that were administered in both orders (Fig. 2). Four homologous prime-boost control groups were primed and boosted with the same vaccine strain, and a mock-vaccinated control group was injected twice with phosphate-buffered saline (PBS) in combination with adjuvant.

We first examined the evolution of functional antibody titers against the four vaccine strains and both strains selected for challenge. Pre-vaccination sera were negative in hemagglutination inhibition (HI), virus neutralization (VN) and neuraminidase inhibition (NI) assays. Mock-vaccinated control pigs tested negative for HI antibodies (<10) but occasionally had minimal VN (4) or NI (≤40) titers. We consider HI titers ≥40 and VN titers ≥64 as seroprotective, based on the accepted seroprotective threshold for seasonal influenza in humans[21]. In influenza vaccine naïve children, on the other hand, HI titers of 330 were associated with 80% protection[22]. Likewise, HI titers ≥640 and VN titers ≥1024 were needed to ensure sterilizing immunity in vaccination-challenge studies in pigs[15].

Post-vaccination HI and VN antibody titers followed a similar pattern, but VN titers were higher. One month after the first vaccination, antibodies were either undetectable or detectable against the respective vaccine strains only, with mean HI titers ≤40 and mean VN titers ≤64. Antibody titers peaked 14 days after the second vaccination and were lower at 28 days, before challenge. At that time, the homologous prime-boost groups had seroprotective HI and VN titers against the strain used for vaccination, but not against the other three vaccine strains (Table 1). G10 tended to induce lower homologous antibody titers than the other vaccine strains, which may be due to a lower immunogenicity and/or a lower antigen dose in the vaccine. The heterologous prime-boost groups were also largely lacking antibodies against the two vaccine strains to which the pigs had not been exposed, but five out of eight groups had seroprotective HI and/or VN titers against both vaccine strains administered, compared to only one strain in the homologous prime-boost controls. By contrast, two groups (G10-IL05, G10-ARM08) had HI and VN titers below the seroprotective threshold against all strains. All vaccinated groups had seroprotective titers against one of both challenge strains at best.

Post-vaccination NI antibody titers followed similar kinetics as HI/VN titers, but they were higher and showed broader cross-reactivity (Table 1). One month after homologous prime-boost vaccination, three of the four groups had NI titers ≥160 against the vaccine strains only. This contrasted with the eight heterologous prime-boost groups, six of which had titers ≥160 against both strains used for vaccination and, with one exception, one or both remaining vaccine strains. The CA09-CA09 and G10-CA09 groups had such titers against both challenge strains. In all three assays, antibody titers were generally highest against the strain used for the primary vaccination and reversing the vaccine strain order could change antibody profiles, as shown by the ARM08-IL05 and IL05-ARM08 groups.

Thus, sequential administration of two heterologous vaccine strains may stimulate functional antibodies against both strains. The cross-reactivity and magnitude of the antibody response varies between different prime-boost regimens and appears to depend on factors such as the order of administration.

### Breadth of H1N1 antibodies after two-dose vaccination

To compare the breadth of the antibody response between vaccine groups, sera collected 2 weeks after the second vaccination were pooled for each group and examined in HI/VN assays against 24 antigenically diverse H1 virus strains and in NI assays against 14 N1 strains. The test strains included the vaccine and challenge strains, additional

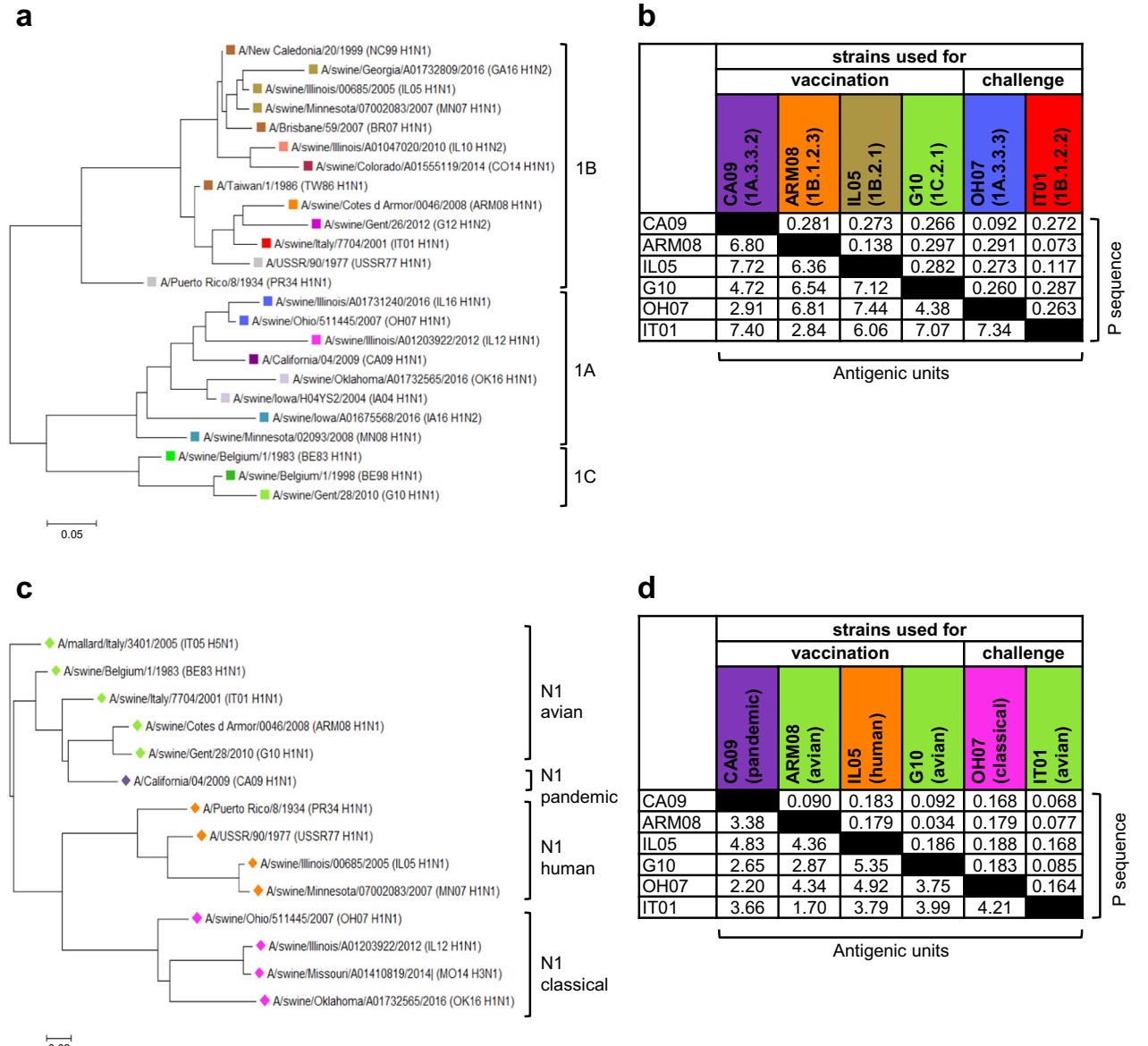

**Fig. 1 | Genetic and antigenic relationships between influenza A virus strains.**
**a**, **c** Phylogenetic trees based on amino acid (aa) sequences of the HA1 (**a**) and NA (**c**) of all H1 and N1 strains used. Phylogenetic relationships were estimated using the maximum-likelihood method in MEGA version 7.0 software and the Jones-Taylor-Thornton substitution model. Branch length is proportional to genetic distance. The scale bar indicates aa substitutions per site. Colors were used to indicate H1 lineages (1A, 1B, 1C) and clades, as described by Anderson et al.[18], and N1 lineages (avian, pandemic, human, classical). **b**, **d** Genetic and antigenic distances between the HA1 (**b**) and NA (**d**) proteins of influenza A virus (IAV) strains used for vaccination and challenge. Genetic distances are expressed as P sequence values (upper right triangle). P sequence is defined as: Number of aa substitutions in the HA1 domain of HA / Total number of aa in the HA1 domain of HA[56]. Antigenic distances are expressed in antigenic units (lower left triangle), which were calculated as described by Peeters et al[59]. See Supplementary Tables 2, 4, 5 and 7 for comparisons between all H1 and N1 strains used.

swine IAV strains from Europe or North America, historical human strains, and one avian strain (Fig. 3). Supplementary Tables 2–7 demonstrate the genetic and antigenic relationships between the HA and NA of all strains. Mock-vaccinated control groups tested negative in all three assays (data not shown). The breadth and magnitude of the antibody responses of the vaccinated groups are shown in heatmaps.

HI and VN antibody responses of the homologous prime-boost groups remained mostly lineage-specific (Fig. 3a, b). Only two heterologous prime-boost groups, CA09-G10 and CA09-ARM08, had broader antibody responses than the matched homologous control groups (Supplementary Table 8). Still, there was no enhancement of cross-lineage reactivity and both groups were largely lacking seroprotective titers against 1B lineage swine H1 virus strains from North

America and human H1N1 strains. In addition, antibody titers did not exceed those of the CA09-CA09 group.

NI assays showed a lack of cross-reactivity between swine IAV strains with a human-origin N1 and those with an N1 of avian, pandemic or classical swine origin (Fig. 3c). Again, the heterologous prime-boost groups mainly reacted with the same strains as the corresponding homologous prime-boost groups and their antibody profiles at best resembled a combination of those of the homologous control groups. All groups had NI titers <160 against the historical human strains A/Puerto Rico/8/1934 (PR34) and A/USSR/90/1977 (USSR77).

In short, two administrations of heterologous monovalent vaccine failed to induce a pan-H1 or pan-N1 antibody response.

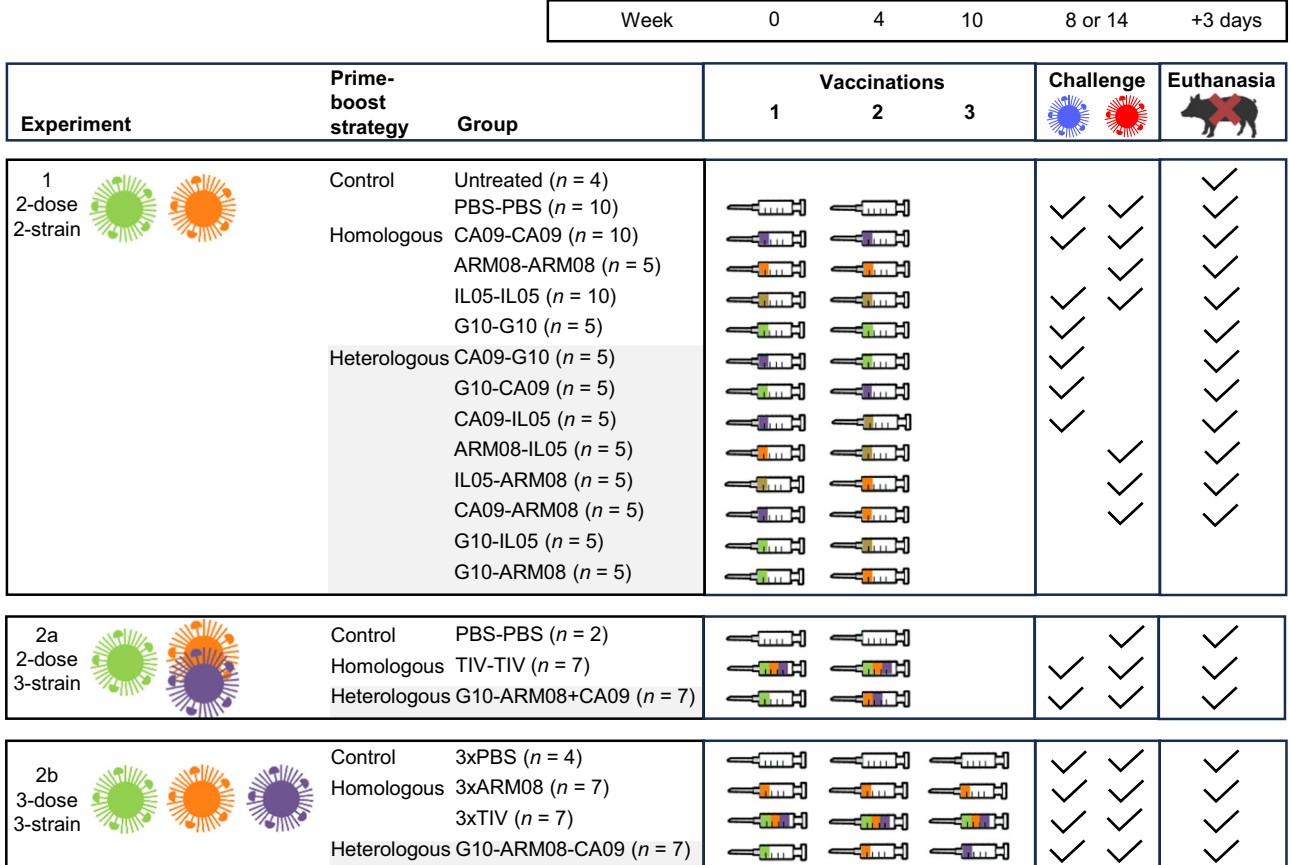

**Fig. 2 | Experimental design of the study.** Three separate experiments (1, 2a, 2b) were performed. The viruses illustrate (one of) the heterologous prime-boost vaccination regimen(s) of each experiment. The colors of viruses and within syringes refer to vaccine strains from different H1 clades: CA09 (1A.3.3.2, violet), ARM08 (1B.1.2.3, orange), IL05 (1B.2.1, khaki), G10 (1C.2.1, green). Pigs were given two vaccinations with 4 weeks interval (experiment 1 and 2a), and a third vaccination 6 weeks after the second (experiment 2b). Heterologous prime-boost groups (grey shading) received different vaccine strains for the first and second and/or third vaccination. They were compared with homologous prime-boost groups and mock-vaccinated control groups (PBS). Except for two vaccinated groups and an untreated control group in experiment 1, all groups were challenged one month after the last vaccination. Both challenge virus strains were antigenically distinct from the strains used for vaccination and their colors also refer to H1 clades: OH07 (1A.3.3.3, blue), IT01 (1B.1.2.2, red). Pigs were euthanized 3 days post challenge to determine virus titers in the respiratory tract. See Supplementary Table 1 for full names of virus strains. TIV trivalent vaccine.

## Antibodies and secreting cells after three-dose vaccination

In a second experiment, we tried to induce broader and higher antibody responses by heterologous prime-boost vaccination with IAV strains from three different H1 lineages. A two-dose heterologous prime-boost group was primed with G10 and boosted with bivalent ARM08 + CA09 vaccine (experiment 2a), and a three-dose group was vaccinated sequentially with these three strains (experiment 2b). Control groups received two or three doses of homologous trivalent (TIV) vaccine, or three doses of monovalent ARM08 vaccine. Apart from the evolution of antibody titers against the vaccine and challenge strains, we also determined the frequency of IgG antibody-secreting cells (ASC) against these strains in the circulation. To do so, we performed enzyme-linked immunospot (ELISpot) assays with purified whole virus and peripheral blood mononuclear cells (PBMC) collected at the time of the second and/or third vaccination and 7 days later. The 7-day time point was selected because plasmablast responses have been shown to peak one week after immunization of primed subjects[8].

The mock-vaccinated control pigs tested largely negative in serological and ELISpot assays throughout the experiment. All vaccinated groups had minimal serological responses after the first vaccination. Most pigs of the G10-ARM08-CA09 group also had HI and VN titers below the seroprotective threshold after the second vaccination (Fig. 4c), like the G10-ARM08 group in experiment 1. One month after the final vaccination, the seroprotective threshold was reached against all three vaccine strains in the TIV-TIV, 3xTIV and G10-ARM08-CA09 groups, but not in the remaining two groups (Table 1). The former groups had seroprotective titers against both challenge strains, but not against IL05. The 3xTIV and G10-ARM08-CA09 groups also had NI titers ≥160 against all three vaccine strains and both challenge strains, unlike the other groups.

Virus-specific ASC were undetectable at the time of the second and/or third vaccination (data not shown) but increased to group means of 1–30 per million PBMC one week after the booster vaccination(s). Because ASC may target either one of both surface proteins, we have depicted ASC numbers at day 7 along with VN and NI titers at days 0 and 28 post vaccination (Fig. 4). ASC numbers tended to be highest against the strains used for vaccination and lowest against the IT01 challenge strain, which was most distant from the vaccine strains. There was a large degree of inter-animal and inter-assay variability, as observed by the differences after the second TIV dose in the two- and three-dose groups. The two-dose heterologous prime-boost group (Fig. 4b) had slightly higher ASC numbers than the TIV-TIV group, but roughly similar antibody titers at day 28. The three-dose heterologous prime-boost group (Fig. 4c) showed some obvious differences with the homologous prime-boost groups. The former group still had 1.5- to 2-fold lower ASC numbers than the 3xTIV group after the second vaccination, and lower antibody titers against CA09 and/or ARM08. Importantly, the third dose of heterologous vaccine was followed by 2-

**Table 1 | Pre-challenge antibody titers against vaccine and challenge (Ch) strains in hemagglutination inhibition (HI), virus neutralization (VN) and neuraminidase inhibition (NI) assays**

| Experiment | Prime-boost | Group | HI Vaccine strains CAO9 (1A.3.3.2) | ARMO8 (1B.1.2.3) | ILO5 (1B.2.1) | G1O (1C.2.1) | HI Ch strains OHO7 (1A.3.3.3) | ITO1 (1B.1.2.2) | VN Vaccine strains CAO9 (1A.3.3.2) | ARMO8 (1B.1.2.3) | ILO5 (1B.2.1) | G1O (1C.2.1) | VN Ch strains OHO7 (1A.3.3.3) | ITO1 (1B.1.2.2) | NI Vaccine strains CAO9 (pandemic) | ARMO8 (avian) | ILO5 (human) | G1O (avian) | NI Ch strains OHO7 (classical) | ITO1 (avian) |
|---|---|---|---|---|---|---|---|---|---|---|---|---|---|---|---|---|---|---|---|---|
| 1 | Ctrl. | PBS-PBS | <10 | <10 | <10 | <10 | <10 | <10 | <4 | <4 | <4 | <4 | <4 | <4 | 15 | <10 | 12 | <10 | 25 | <10 |
| | Hom. | CAO9-CAO9 | 452* | 6 | <10 | 9 | 184 | 6 | 1036* | 35 | <4 | 18 | 386* | 12 | 5581* | 368* | 43 | 422* | 1040* | 172 |
| | | ARMO8-ARMO8 | 6 | 139* | 9 | <10 | 10 | 92 | 6 | 2409* | 9 | <4 | 4 | 1048* | 106 | 970* | 70 | 139 | 35 | 243 |
| | | ILO5-ILO5 | <10 | <10 | 422* | <10 | 15 | <10 | <4 | <4 | 2965* | <4 | 9 | 7 | 61 | 23 | 3620* | 15 | 46 | 12 |
| | | G1O-G1O | 6 | <10 | <10 | 80* | 12 | <10 | 5 | <4 | <4 | 234* | 6 | 4 | 61 | 80 | 13 | 368* | 46 | 12 |
| | Het. | CAO9-G1O | 92* | 6 | 6 | 46 | 106 | <10 | 467* | 6 | <4 | 96 | 247* | 3 | 1280* | 279 | 20 | 485* | 243 | 35 |
| | | G1O-CAO9 | 40 | 6 | <10 | 80* | 80 | <10 | 87 | 7 | <4 | 108 | 111 | 4 | 1940* | 2941* | 30 | 5881* | 279 | 160 |
| | | CAO9-ILO5 | 53 | 8 | 17 | 6 | 53 | 9 | 139* | 17 | 75 | <4 | 69 | 7 | 1689* | 970* | 184 | 1470* | 211 | 106 |
| | | ARMO8-ILO5 | <10 | 160* | 61* | <10 | 6 | 15 | <4 | 1723* | 146 | <4 | <4 | 50 | 40 | 279 | 184 | 53 | 46 | 30 |
| | | ILO5-ARMO8 | <10 | 13 | 26 | <10 | <10 | 12 | <4 | 57 | 124* | 4 | 3 | 22 | 92 | 46 | 160 | 61 | 61 | 17 |
| | | CAO9-ARMO8 | 61 | 17 | <10 | 7 | 20 | 13 | 177* | 117 | <4 | 17 | 19 | 42 | 557* | 243 | 35 | 485* | 139 | 46 |
| | | G1O-ILO5 | 6 | <10 | 26 | 17 | 10 | <10 | <4 | 3 | 60 | 11 | 4 | <4 | 279 | 320* | 139 | 485* | 61 | 23 |
| | | G1O-ARMO8 | <10 | 12 | <10 | 9 | 7 | 7 | <4 | 41 | <4 | <4 | 4 | 8 | 61 | 53 | 17 | 139 | 53 | 10 |
| 2a | Ctrl. | PBS-PBS | <10 | <10 | <10 | <10 | <10 | <10 | <4 | <4 | <4 | <4 | <4 | <4 | 7 | 10 | 7 | 7 | 40 | <10 |
| | Hom. | TIV-TIV | 476* | 66* | 6 | 108* | 98 | 16 | 260* | 305* | <4 | 129* | 73 | 50 | 525* | 390* | 22 | 780* | 320* | 72 |
| | Het. | G1O-ARMO8 + CAO9 | 580* | 8 | <10 | 177* | 195 | 6 | 175* | 30 | <4 | 189* | 181* | 8 | 320* | 1050* | 18 | 2319* | 476* | 66 |
| 2b | Ctrl. | 3xPBS | <10 | <10 | <10 | <10 | <10 | <10 | <4 | <4 | <4 | <4 | <4 | <4 | <10 | <10 | <10 | <10 | <10 | <10 |
| | Hom. | 3xARMO8 | 16 | 320* | 8 | <10 | 11 | 431 | 3 | 1218* | 11 | <4 | <4 | 1291* | 59 | 320* | 20 | 263 | 44 | 290* |
| | | 3xTIV | 861* | 263* | 12 | 476* | 476 | 320 | 500* | 999* | 8 | 552* | 684* | 854* | 1902* | 476* | 36 | 1902* | 1413* | 431* |
| | Het. | G1O-ARMO8-CAO9 | 2319* | 238* | 7 | 525* | 580 | 476 | 2478* | 2851* | 7 | 1077* | 1016* | 1485* | 4200* | 12483* | 72 | 22612* | 1902* | 1280* |

Pigs were given two vaccinations with 4 weeks interval (experiment 1 and 2a), and a third vaccination 6 weeks after the second (experiment 2b). Antibody titers were determined 28 days after the final vaccination, before challenge. Ctrl.: mock-vaccinated control group; Hom.: homologous prime-boost groups; Het.: heterologous prime-boost groups. Geometric mean antibody titers of each group are given. Numbers of pigs in each group are shown in Fig. 2. Sera were tested at an initial dilution of 1:10 (HI, NI) or 1:4 (VN). Bold font indicates HI titers ≥40 and VN titers ≥64, which is the accepted seroprotective threshold in humans, and NI titers ≥160. Asterisks indicate HI and VN titers that are statistically higher (two-sided $p < 0.05$, Generalized Linear Model with y-intercept suppressed and threshold subtracted from log-transformed titers) than the seroprotective threshold, and NI titers that are statistically higher than 160. A Source Data file provides source data and includes individual antibody titers, geometric means, and standard deviations.

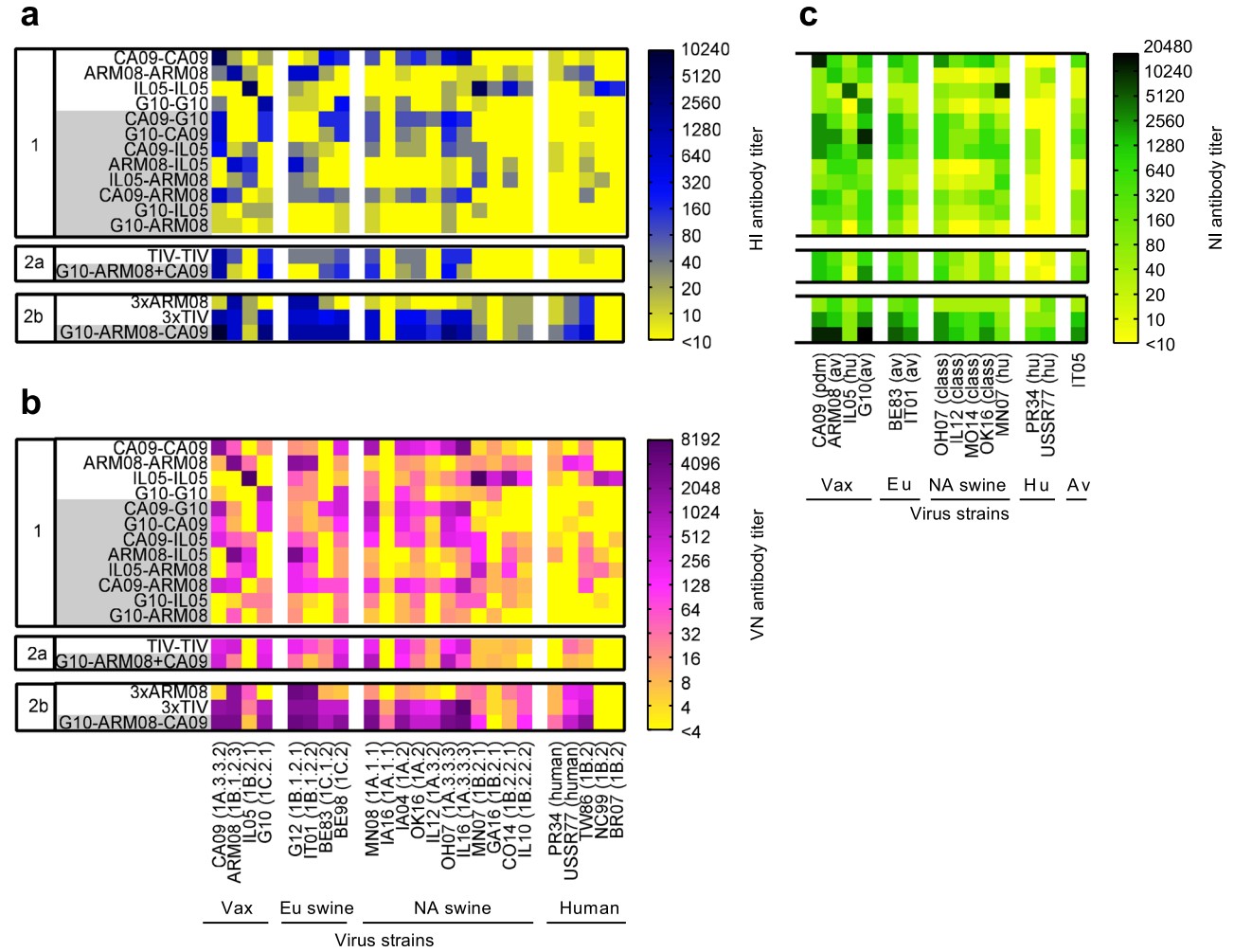

**Fig. 3 | Breadth of functional antibody responses. a** HI antibody titers against a panel of 24 H1 virus strains. **b** VN antibody titers against the same 24 H1 strains. HA lineages and clades[18] are shown between brackets. **c** NI antibody titers against a panel of 14 N1 virus strains. These have an N1 derived from the 2009 pandemic (pdm), Eurasian avian (av), classical swine (class), or human seasonal (hu) H1N1 virus lineages. IT05 is a wholly avian H5N1 virus strain. The vaccine strains (Vax) are shown first, followed by swine IAV strains from Europe (Eu) and North America (NA) and human (hu) and/or avian (av) virus strains. See Supplementary Table 1 for full virus names. Pigs were given two vaccinations with 4 weeks interval (experiment 1 and 2a), and a third vaccination 6 weeks after the second (experiment 2b). Each row shows antibody titers against the various strains in pooled serum from a given homologous (white background) or heterologous (grey background) prime-boost group, 14 days after the final booster vaccination. Mock-vaccinated control groups tested negative in all assays and are not shown. Antibody titers are color coded as shown by the spectrum next to the heatmaps. Source data are provided as a Source Data file.

to 5-fold higher numbers of ASC ($p = 0.0046$–$0.75$, Kruskal–Wallis test) and 10- to 100-fold higher antibody titers as compared to the values observed after the second dose. In the three-dose homologous prime-boost groups, on the contrary, ASC numbers were comparable after the second and third vaccination ($p = 0.09$–$0.99$) and serum antibody titers showed no or ≤6-fold increases.

Finally, G10-ARM08-CA09 was the single group with ≥20 ASC per million PBMC against four of the five strains tested and VN and NI titers ≥1000 against all five strains, including both challenge strains (Table 1). This suggests that a third dose of heterologous monovalent vaccine primes pigs for stronger plasmablast and antibody responses.

### Breadth of H1N1 antibodies after three-dose vaccination
Next, we examined pooled sera from experiment 2 for their reactivity with the HI/VN and NI virus panels. The two-dose vaccine groups failed to reach higher cross-reactivity or total titer scores than the best performing groups of experiment 1 (Fig. 3). The 3xTIV and G10-ARM08-CA09 groups, by contrast, had higher HI, VN and NI titers than any two-dose group. The latter group also had seroprotective HI and VN titers against some of the North American 1B lineage swine H1 virus strains.

Similarly, three doses of heterologous vaccine resulted in NI scores ≥160 against 13 of 14 N1 strains, including the historical human strains PR34 and USSR77.

In all three assays, the G10-ARM08-CA09 group showed greater serologic cross-reactivity than any other group, including the 3xTIV group (Supplementary Table 8). The three-dose heterologous prime-boost group had seroprotective VN titers against 71% of a diverse panel of H1 virus strains, some of which were more than 5–6 antigenic units away from the vaccine strains, and detectable VN titers against 88% of these strains. Thus, three doses of heterologous monovalent vaccine induce a near-pan H1N1 antibody response.

### Further analysis of anti-HA and -NA antibodies
We then examined pre-challenge serum pools from each group for functional antibodies against swine and avian IAV strains with heterosubtypic HA and/or NA proteins (Fig. 5a, b). As expected, most vaccinated groups tested positive in NI assays with avian H5N1 and H7N1 strains, confirming the broad reactivity of anti-N1 antibodies. HI antibodies were detected against both H5N1 strains, but they were mostly lacking against H5N2, H5N9 or non-H5 strains. We failed to

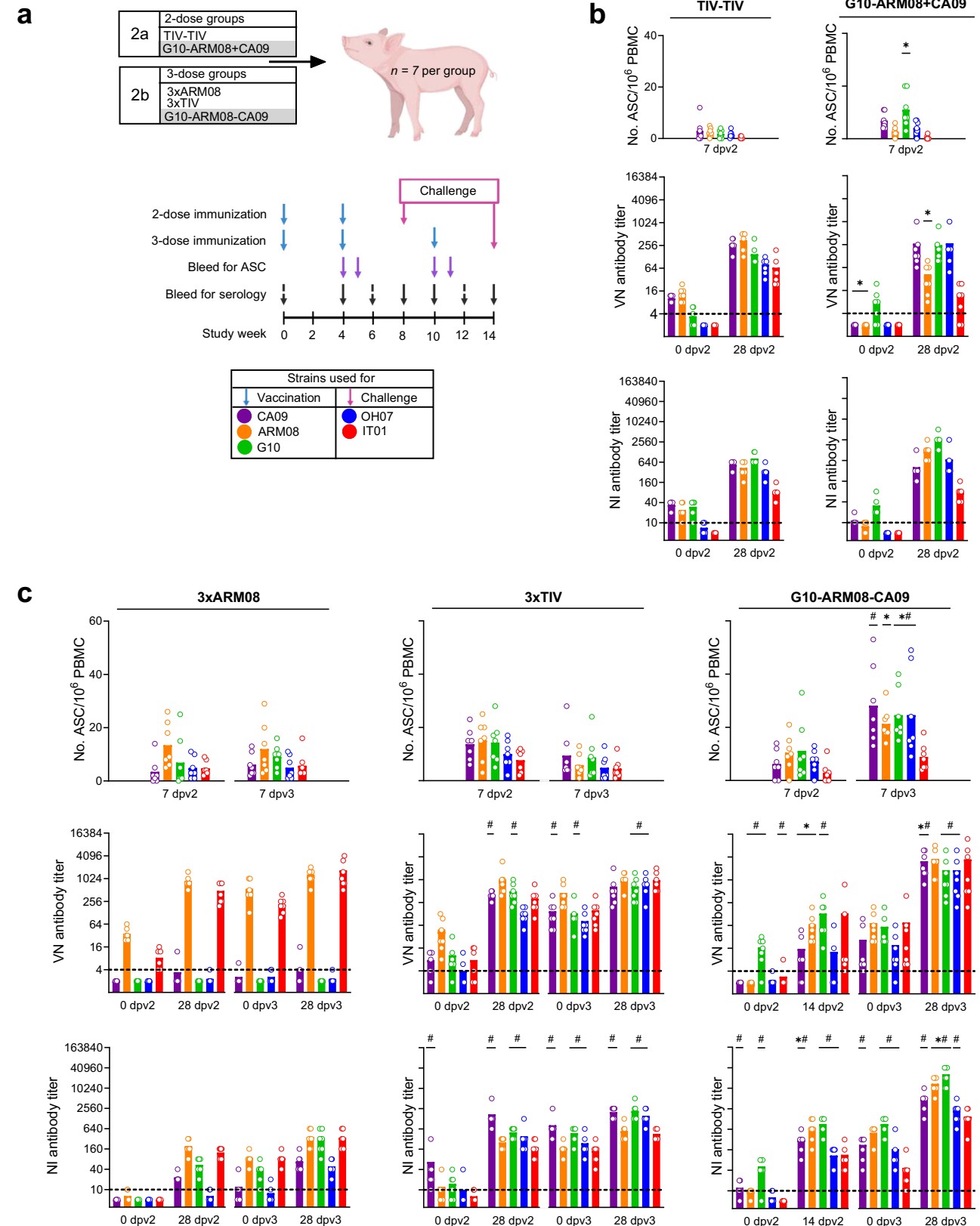

detect VN antibodies against any strain but found a strong positive correlation (Spearman's $\rho = 0.96$ for MN81 and $0.94$ for IT05, $p < 0.0001$) between H5N1 antibody titers in HI and NI assays.

We also tested individual pre-challenge sera by enzyme-linked immunosorbent assay (ELISA) for antibodies against the conserved HA stalk and nucleoprotein (NP). Mean IgG ELISA titers against both proteins were higher in three-dose than in two-dose vaccine groups (Fig. 5c, d), but there were no differences between homologous and heterologous prime-boost groups, and no associations with VN antibody titers. These results indicate that the neutralizing antibody response does not extend beyond the H1 subtype and mainly targets the head of the HA.

**Fig. 4 | Antibody-secreting cells and evolution of VN and NI antibody titers.**
**a** Schematic of the prime-boost regimens in experiment 2. Pigs were vaccinated with two (experiment 2a) or three doses (experiment 2b) of homologous or heterologous vaccine and challenged one month after the final vaccination. Peripheral blood mononuclear cells (PBMC) for ELISpot assays were collected at the time of each vaccination and 7 days later. Blood for serological assays was collected every two weeks, except for week 2. The three vaccine strains (CA09, ARM08, G10) and both challenge strains (OH07, IT01) were used as test antigens. **b**, **c** Numbers of virus-specific antibody-secreting cells (ASC) 7 days after each booster vaccination

and VN and NI antibody titers at 0 and 28 days after each booster. In the G10-ARM08-CA09 group, titers were determined at 14 days post vaccination 2, because sera from day 28 were not available. Group mean (bars) and individual values (dots, $n = 7$ per group) are shown. The two-dose (**b**) and three-dose (**c**) heterologous prime-boost groups were compared with the matched homologous prime-boost group(s). Asterisks (*) denote statistically significant differences with the matched TIV groups and number signs (#) denote differences with the 3xARM08 group (*$p < 0.05$, #$p < 0.05$, Kruskal–Wallis test). Source data are provided as a Source Data file.

We then tried to further dissect the antibody response against the HA head by HI assays with a panel of five recombinant H1 viruses expressing the H1 protein of H1N1pdm09 strain A/Michigan/2015 (MI15) and carrying aa substitutions in each of the five classically defined antigenic sites. The fold reduction in HI titer against a given antigenic site as compared to the titer against wild type (wt) virus, or the dominance index, is a measure for the contribution of that antigenic site to the overall HI antibody response. To measure the HI activities of non-classical epitopes, we used a mosaic H5/1 (mH5/1) virus, in which all five classical H1 antigenic sites were replaced with exotic sequences from an H5 HA, leaving non-classical epitopes intact.

HI titers against the wt MI15 virus and its antigenic site mutants were only detectable in pigs vaccinated with CA09, which differs from MI15 in four aa in three antigenic sites (Fig. 6b, Table 2). The CA09-CA09 and TIV-TIV groups had reduced HI titers against H1-ΔSb and H1-ΔCa2 (Fig. 6a). Reductions in HI titers against H1-ΔSa were observed in some of the two-dose heterologous prime-boost groups, especially in the G10-ARM08 + CA09 group. In the three-dose groups, HI antibodies did not seem to be directed against any specific antigenic site. Interestingly, three of the seven pigs of the G10-ARM08-CA09 group had HI titers ≥40 against the mH5/1 virus, whereas all other pigs had titers ≤10.

Thus, antigenic sites Sb and Ca2 seem to be immunodominant after homologous prime-boost vaccination with CA09 or TIV, but heterologous prime-boost vaccination may shift the antibody response to other sites. Antibody responses against non-classical HA head epitopes are uncommon but can be boosted by a third dose of a heterologous vaccine strain.

### Protection against challenge with heterologous H1 clades
Finally, we evaluated virological protection against heterologous H1N1 swine IAV strains of the classical 1A (A/swine/Ohio/511445/07, OH07) or human-like 1B (A/swine/Italy/7704/01, IT01) H1 lineage. Comparisons of antigenic sites with the most closely related vaccine strains showed six aa differences between OH07 and CA09 and nine aa differences between IT01 and ARM08. One month after the final booster vaccination, pigs were challenged intranasally with 7.0 $\log_{10}$ tissue culture infectious doses (TCID$_{50}$) of virus. Both challenge strains replicated to high virus titers in mock-vaccinated control pigs (Fig. 7).

Most two-dose vaccine groups had reduced mean virus titers after challenge with OH07, but CA09-CA09 was the single group with significant reductions in all samples of the respiratory tract. Similarly, the ARM08-ARM08 group was best protected against challenge with IT01, while other two-dose groups had reduced virus titers in the lower respiratory tract only or no protection at all. Three doses of ARM08 vaccine offered complete protection against IT01 challenge, except for a tracheal sample of one pig, but minimal protection against OH07. By contrast, the 3xTIV and G10-ARM08-CA09 groups showed a near-sterile protection against both virus strains. There was a strong negative correlation between virus titers and HI, VN and NI antibody titers against the challenge strains ($\rho = -(0.69-0.88)$, $p < 0.0001$). The correlation between virus titers and antibody titers against the H1 stalk ($\rho = -(0.34-0.65)$, $p = 0.0001-0.081$) and NP ($\rho = -(0.39-0.54)$, $p = 0.0001-0.016$) was much weaker.

We also assessed lesions of the lungs and trachea post challenge. Unvaccinated, unchallenged control pigs, which tested negative for swine IAVs and antibodies until the end of the study, did not show gross lung lesions or microscopic lesions of the trachea. Lung alveolar damage and peribronchiolar lymphocytic cuffing were occasionally seen, resulting in a mean microscopic lung lesion score of 2. Challenge of mock-vaccinated control pigs with OH07 or IT01 resulted in minimal macroscopic lung lesions, involving 0 to 8% of the lungs of individual pigs. Microscopic lesions were mild and consisted of accumulations of lymphocytes and neutrophils in or around bronchioles, bronchiolar epithelial damage, thickening of the alveolar walls and focal attenuation of the tracheal epithelium (Supplementary Table 9). Most vaccinated groups had negligible or mild pathology in comparison with the control groups, but the differences were rarely significant due to limited sample sizes and individual variation. The IL05-IL05 group stood out because of more severe macroscopic and microscopic lung lesions than the unvaccinated controls with either one of both challenge viruses. These lesions could be regarded as "vaccine-associated enhanced respiratory disease" (VAERD), which may occur in pigs vaccinated with adjuvanted WIV influenza vaccine with the HA as well as NA proteins mismatched from the challenge virus strain[23]. However, the differences with the mock-vaccinated challenge control groups remained minor ($p = 0.05$ for microscopic lung lesion score after IT01 challenge, $p > 0.99$ for other scores, Kruskal-Wallis). There were only moderate correlations between lesion scores and viral loads in the lungs ($\rho = 0.36$ for OH07, 0.06 for IT01) or trachea ($\rho = 0.62$ for OH07, 0.14 for IT01) of individual pigs.

In summary, the broad and high anti-H1N1 antibody responses in the G10-ARM08-CA09 group translated into near-complete protection against both classical swine and human-like H1N1 swine IAV. Three doses of heterologous monovalent vaccine were as effective as three doses of matched TIV vaccine and better than two doses, while the total amount of antigen per dose was 3-fold lower.

## Discussion
Influenza vaccines that protect against any H1N1 IAV strain from swine or humans would be a breakthrough for both species. Heterologous prime-boost vaccination with antigenically distinct H1 antigens is a proven approach to increase the breadth of anti-HA antibodies, as shown in experiments in mice and ferrets[24–26]. Yet these experimental studies focused on the 2009 pandemic and pre-pandemic human seasonal H1N1 strains, and they did not result in a pan-H1N1 neutralizing antibody response or protection. Our study is different in that we used the pig model and whole inactivated, adjuvanted virus vaccines based on H1pdm09 and diverse swine IAVs. We tested multiple two- as well as three-dose heterologous prime-boost regimens. Besides, we determined functional antibody titers against a large panel of H1 IAV strains from humans and swine, spanning multiple decades and geographic regions. As in previous studies, most two-dose heterologous prime-boost regimens stimulated HI and VN antibodies against both vaccine strains, but they failed to induce a pan-H1 antibody response. In contrast, three sequential administrations of heterologous monovalent H1N1 vaccine induced seroprotective neutralizing antibodies against 71% of the H1 virus panel, and

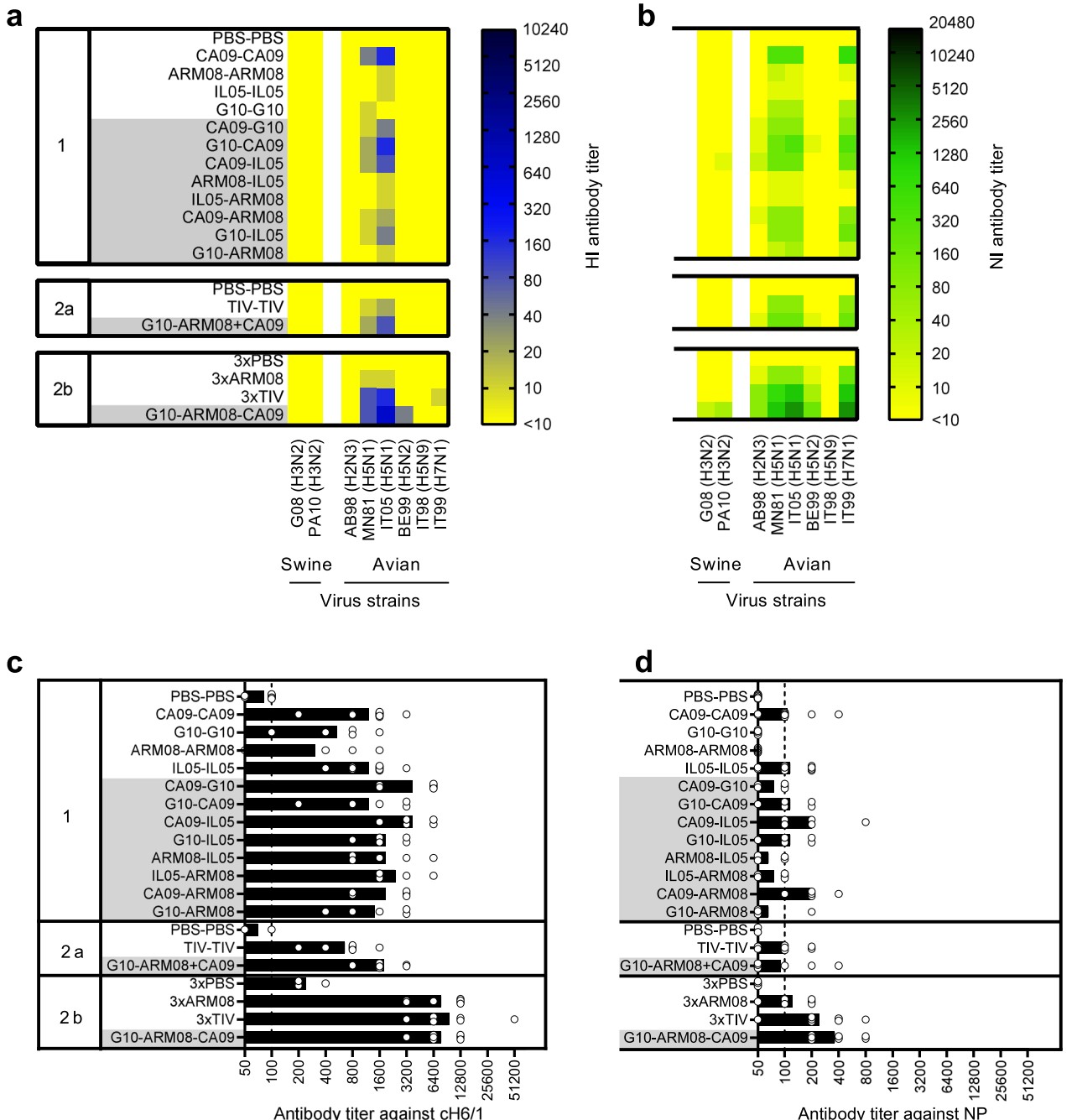

**Fig. 5 | Antibody titers against heterosubtypic influenza A virus strains, the H1 stalk and the nucleoprotein. a** HI antibody titers against two swine and six avian influenza A virus (IAV) strains. **b** NI antibody titers against the same strains. Subtypes are shown between brackets. The HA and NA of the test strains belong to phylogenetic group 1 (H2, H5; N1), as for the H1N1 vaccine strains, or to group 2 (H3, H7; N2, N3, N9). See Supplementary Table 1 for full virus names. Each row shows antibody titers against the various strains in pooled serum from a given homologous (white background) or heterologous (grey background) prime-boost group of experiment 1, 2a and 2b, at 28 days after the final booster vaccination. Antibody titers are color coded as shown by the spectrum next to the heatmaps. All samples

tested negative in VN assays. **c** IgG ELISA titers against a recombinant chimeric cH6/1 HA antigen with an H1 stalk domain from A/California/04/2009 (CA09). **d** IgG ELISA titers against a recombinant nucleoprotein (NP) derived from A/Michigan/45/2015 (MI5). Group mean (bars) and individual antibody titers (dots) are shown. Numbers of pigs in each group are shown in Fig. 2. The dotted lines indicate the detection limit. Both anti-HA stalk and anti-NP titers were higher in the three-dose than in the two-dose groups (p = 0.0029 and 0.0088 respectively, two-sided Mann–Whitney test), but there was no difference between homologous and heterologous prime-boost groups (p = 0.40 and 0.80, two-sided Mann–Whitney test). Source data are provided as a Source Data file.

detectable neutralizing antibodies against 88%. This shows that even the most traditional influenza vaccines can offer a surprisingly broad protection if they are administered in a novel way. Our study affirms recent proposals that sequential immunizations with different HA proteins boost responses to those epitopes that are shared by multiple

virus strains and thus maximize the broadly neutralizing repertoire against influenza viruses[1,2,27].

One of the most remarkable findings was the rapid and robust expansion of influenza virus-specific ASC in the circulation after the third administration of heterologous vaccine. This contrasted with

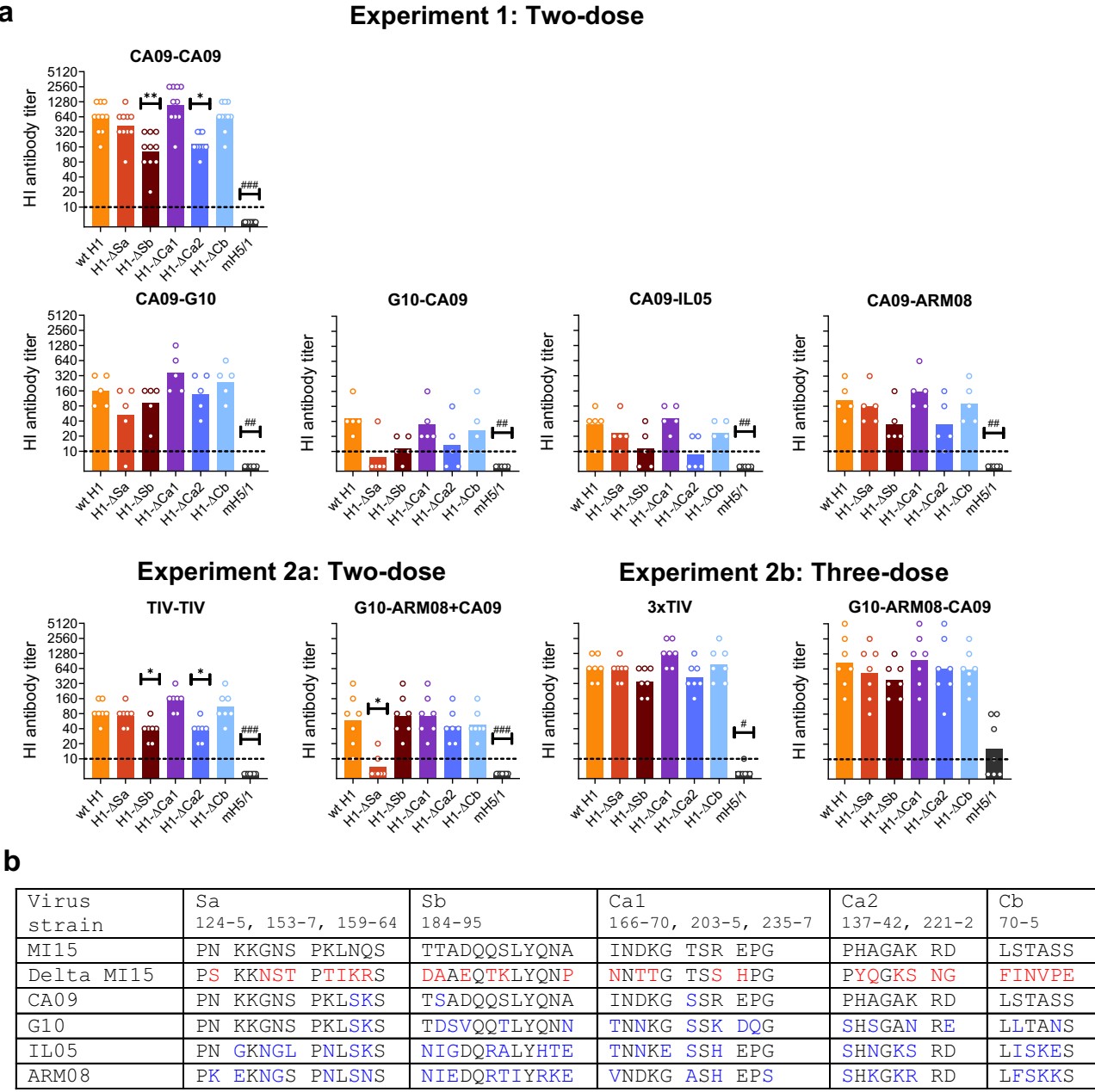

**Fig. 6 | HI profiles against antigenic sites in the H1 head. a** HI antibody titers against a panel of five recombinant H1 viruses (H1-Δ) expressing the H1 protein of H1N1pdm09 strain A/Michigan/45/2015 (MI15, wt H1) and carrying aa substitutions in one of the five classically defined antigenic sites (Sa, Sb, Ca1, Ca2, Cb). HI titers were also determined against a mosaic H5/1 virus (mH5/1), in which all five antigenic sites were replaced with those of an H5 influenza virus. Group mean (bars) and individual titers (dots, *n* = 5 or 7 pigs per group, see Table 2) are shown. The dotted lines indicate the detection limit. HI titers against wt H1 were only detectable in groups vaccinated with CA09, and titers against both viruses showed ≤4-fold differences (*p* = 0.10, two-sided Mann-Whitney test; data not shown). Results of other vaccine groups are not shown. Asterisks denote statistically significant reductions in HI titers against a given antigenic site mutant as compared to the titer against wt H1 (*p* < 0.05, **p* < 0.01, Kruskal–Wallis test). The fold reduction is proportionate with the contribution of that antigenic site to the overall HI response. Number signs indicate significantly higher HI titers against mH5/1 in the G10-ARM08-CA09 group than in the other groups (#*p* < 0.05, ##*p* < 0.01, ###*p* < 0.001, Kruskal–Wallis test with Dunn's correction). **b** Alignment showing aa sequences in the five antigenic sites of MI15, and aa differences with each antigenic site of the five mutants (in red) and with the four H1 vaccine strains (in blue). There are four aa differences between MI15 and the prototype H1N1pdm09 strain CA09, compared to 19–28 differences between MI15 and the other strains. Source data are provided as a Source Data file.

relatively minor increases in ASC numbers and serum antibody titers after a third dose of homologous vaccine. Our ELISpot analyses of the ASC response have several limitations. We have only examined lymphocytes in peripheral blood and did not characterize the cells or their target proteins. Also, we tested just a few vaccine groups and time points. Discrepancies between the magnitude of the ASC and serum antibody responses were common. Despite these shortcomings, our

data support the concept that boosting with distinct antigens may recruit broadly reactive memory B cells more efficiently than immunization with identical or similar antigens[27].

As expected, NI antibodies were more cross-reactive than HI or VN antibodies, and they were boosted in an independent manner by heterologous H1N1 vaccine strains[27,28]. This resulted in extraordinarily high NI antibody titers and cross-reactivity between diverse N1 NAs,

**Table 2 | HI dominance indexes against each mutant virus in different vaccine groups**

| Experiment | Prime-boost | Group | N pigs | Average HI dominance index | | | | |
|---|---|---|---|---|---|---|---|---|
| | | | | H1-ΔSa | H1-ΔSb | H1-ΔCa1 | H1-ΔCa2 | H1-ΔCb |
| 1 | Hom. | CA09-CA09 | 10 | 1.5 | 4.8 | 0.6 | 3.8 | 1.0 |
| | Het. | CA09-G10 | 5 | 4.8 | 2.0 | 0.5 | 1.2 | 0.7 |
| | | G10-CA09 | 5 | 6.4 | 4.8 | 1.4 | 4.0 | 2.0 |
| | | CA09-IL05 | 5 | 1.6 | 3.6 | 0.8 | 4.8 | 1.6 |
| | | CA09-ARM08 | 5 | 1.4 | 3.2 | 0.7 | 3.2 | 1.2 |
| 2a | Hom. | TIV-TIV | 7 | 1.3 | 2.6 | 0.6 | 2.7 | 0.9 |
| | Het. | G10-ARM08 + CA09 | 7 | 11.1 | 1.1 | 0.9 | 1.5 | 1.4 |
| 2b | Hom. | 3xTIV | 7 | 1.1 | 2.0 | 0.5 | 1.6 | 0.9 |
| | Het. | G10-ARM08-CA09 | 7 | 1.9 | 2.7 | 0.9 | 1.6 | 2.3 |

The HI dominance index represents the fold reduction in HI titer with a given antigenic site mutant as compared to the titer against wt H1 and is proportionate with the contribution of that antigenic site to the overall HI response.

including those of avian H5N1 and H7N1 strains[29]. An unexpected finding was the detection of anti-H5N1 HI titers in groups with high anti-N1 titers, and the strong correlation between both. We believe that these are non-specific reactions due to steric hindrance of the HA by NA-antibodies. This phenomenon has been previously reported, but the reverse situation is better known[30–34]. Our assumption is based on the lack of anti-H5 antibodies in VN assays or in HI assays with H5N2 or H5N9 strains. This indicates that the neutralizing antibody response is unlikely to extend beyond the H1 subtype. This said, the broad cross-reactivity within the N1 subtype might offer partial protection against divergent viruses that express N1 NAs like H5N1, H7N1, H6N1.

Antibodies against the HA stalk are considered the primary mediators of protection across different HA subtypes from the same phylogenetic group, such as H1 and H5, but their levels remained below those associated with HA stalk-mediated protection against heterosubtypic challenge in experimental animals[35,36]. This is consistent with the idea that optimal expansion of antibodies to subdominant HA stalk epitopes requires sequential exposures to HA proteins with heads from distinct HA subtypes[37]. Likewise, the NP is conserved between all IAV subtypes[38], but anti-NP antibody titers remained below those detected in adult humans or influenza-infected mice[39]. Anti-NP antibodies are not neutralizing and their role in protection remains controversial and may be poor[1,27,39]. While antibody titers against both proteins increased after a third vaccination of pigs, they were similar in homologous and heterologous prime-boost groups and only weakly correlated with protection against heterologous H1N1 challenge. All this supports that protection is limited to the H1 subtype and largely mediated by epitopes in the HA head domain.

To further dissect the antibody response against the HA head, we performed HI assays with antigenic site mutants of the H1N1pdm09 strain MI15, which is closely related to CA09. Until now, these mutants had only been tested against sera from humans, mice, ferrets and guinea pigs[6,40]. The single published study on HA epitope recognition in pigs used different methods[41]. The authors considered Sa and Ca as the dominant antigenic sites in H1N1pdm09 infected pigs, though there was also reaction with the Sb site. In the present study, HI antibodies from CA09 vaccinated pigs reacted mainly with sites Sb and Ca2. However, there was a shift towards Sa immunodominance after two-dose heterologous prime-boost vaccination involving CA09 and G10, especially with G10-ARM08 + CA09. This can be explained by the fact that Sa is the single identical antigenic site between both strains. Interestingly, the Sa immunodominance was abolished by three-dose vaccination with the antigenically distant ARM08 strain in between G10 and CA09. Even more important, vaccination with G10-ARM08-CA09 stimulated low levels of antibodies against a mosaic H5/1 virus, which is missing all five H1 antigenic sites. We hypothesize that such antibodies may target epitopes in the receptor-binding pocket and

lateral patch, as described for several broadly neutralizing human mAbs[2,40,42,43]. Our data suggest that three-dose heterologous prime-boost vaccination may bypass the immune system's focus on the most variable regions of the HA and help to elicit broader anti-head neutralizing antibodies[5,8]. These findings call for further investigation of the potency of the anti-mosaic virus antibodies, their target epitopes, and mechanisms of induction.

The 3xTIV and G10-ARM08-CA09 vaccine regimens offered a near sterile protection against challenge with classical (1A) and human-like (1B) H1 swine IAV strains from clades other than the vaccine strains. The serological profile was even broader for the heterologous regimen, which resulted in HI titers ≥640 and VN titers ≥1024 against multiple additional H1 swine IAV clades from all three lineages, including the Eurasian avian lineage. This suggests that protection may extend to the prevailing swine IAV clades, including those with proposed pandemic potential[44]. While some human-like (1B) strains were not covered, this might change upon additional heterologous booster vaccinations.

Many questions remain unanswered. Will this vaccine strategy also work with longer time intervals between the second and subsequent immunizations? How long does protection last? How much antigenic distance between vaccine strains is required? What is the role of the oil-based adjuvant? An important practical consideration is that the proposed approach requires at least three injections at different times, and that neutralizing antibodies are limited to the H1 subtype. On the upside, the heterologous prime-boost principle has already been shown to lend itself to H3N2 swine IAVs and to multivalent commercial swine influenza vaccines based on distinct H1 and H3 strains[28,45,46]. The present study therefore provides further arguments for the use of different vaccines or antigens for repeat vaccinations of sows. Our results cannot be directly extrapolated to humans, in which preexisting immunity will shape the antibody response to subsequent influenza virus exposures. The phenomenon of immune imprinting or "original antigenic sin" is complex and incompletely understood[47,48]. Depending on the context, it may interfere with or potentiate the response to subsequent influenza vaccinations[27,49,50]. Anyhow, the vastly different immune histories in people of different ages may impact the design and implementation of new prime-boost strategies in humans. Furthermore, influenza vaccines and vaccine policies also differ in swine versus humans. The strains for use in human influenza vaccine production are selected each year and generally show minimal antigenic differences with the previous year's strain. Yet, there is growing evidence that such repeated annual vaccinations may ultimately result in a blunted immune response[51]. We therefore believe that heterologous prime-boost vaccination deserves much deeper investigation, not only for influenza A viruses but also for other constantly evolving respiratory viruses, such as influenza B and SARS-coronavirus-2. Pigs are natural hosts for both

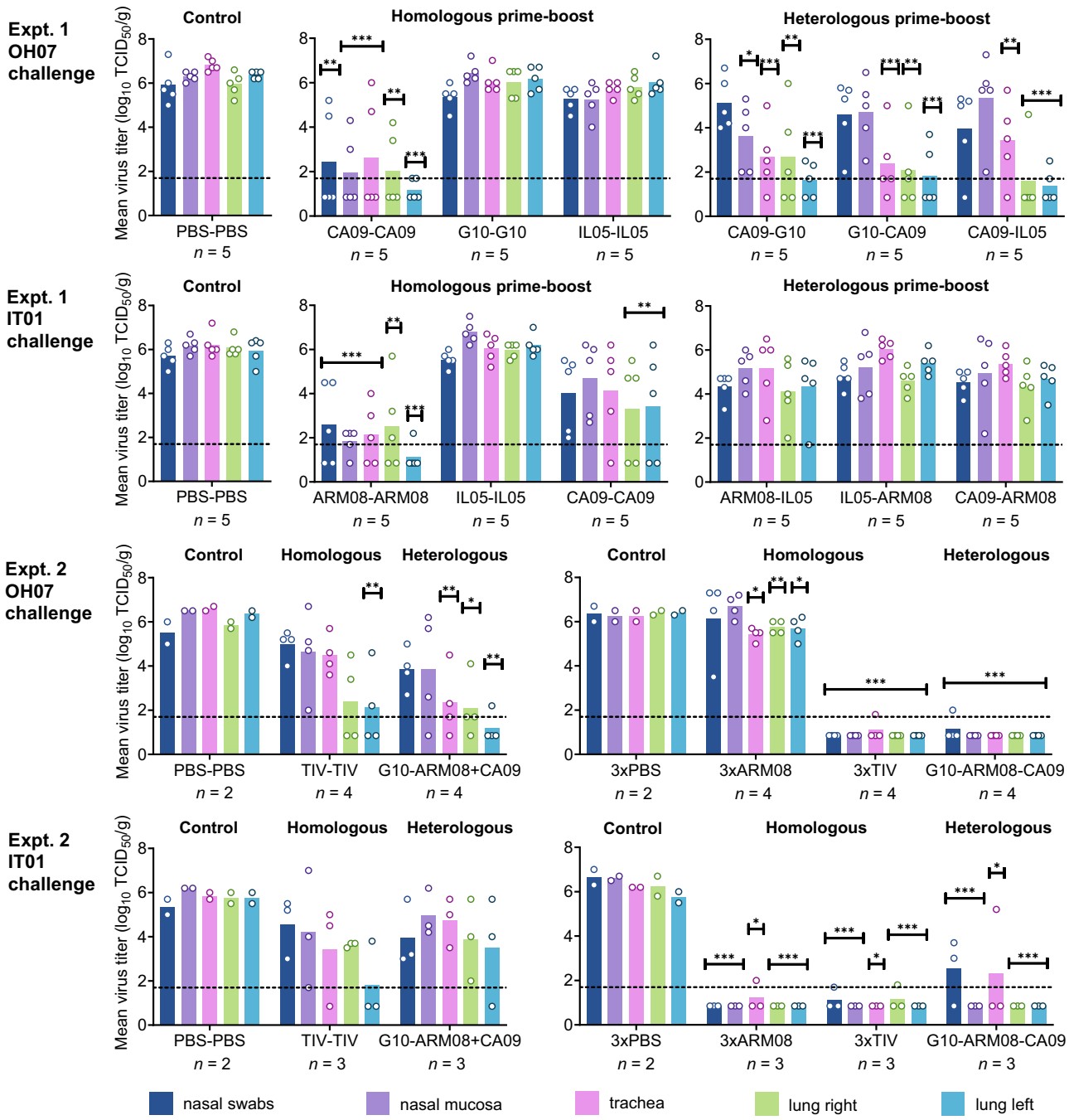

**Fig. 7 | Virus titers post challenge.** Pigs were challenged one month after the final booster vaccination, with H1N1 swine IAV strains of the classical (OH07) or human-like H1 lineage (IT01). They were euthanized 3 days after the challenge, and virus titers were determined in nasal swabs, nasal mucosa, trachea, right and left lung. In experiment 1, virus titers post challenge with OH07 or IT01 are shown in separate graphs for the mock-vaccinated control groups, homologous prime-boost groups and heterologous prime-boost groups. In experiment 2, the two-dose (2a) and three-dose (2b) vaccine groups are shown in separate graphs. Group mean (bars) and individual (dots) virus titers are shown. Dotted lines indicate the limit of detection. Because of a shortage of pigs for challenge with OH07 in experiment 2a, the TIV-TIV and G10-ARM08 + CA09 groups were compared with two randomly selected mock-vaccinated challenge control pigs from experiment 1. Asterisks indicate significantly reduced virus titers as compared to mock-vaccinated challenge control groups (*$p < 0.05$, **$p < 0.01$, ***$p < 0.001$, one-way ANOVA followed by Dunnett's test). Source data are provided as a Source Data file.

influenza and coronaviruses and an excellent animal model to study the immune response to sequential vaccinations[20,52], as well as the effect of prior immunity[53].

## Methods
### Study design and evaluation of antibodies and protection
We aimed to examine whether heterologous prime-boost vaccination with antigenically distinct whole inactivated influenza virus vaccines

can be effective against past and present H1N1 IAV strains from swine and humans. For this purpose, we used the pig model of influenza and experimental vaccines based on four H1N1 IAV strains: A/California/04/2009 (CA09), A/swine/Cotes d'Armor/0046/2008 (ARM08), A/swine/Illinois/00685/2005 (IL05) and A/swine/Gent/28/2010 (G10).

We performed two experiments using a total of 125 4-week-old IAV-naïve crossbred (♀Large White x Landrace) x ♂Piétrain) pigs sourced from a commercial farm (Fig. 2). We used pigs of both sexes:

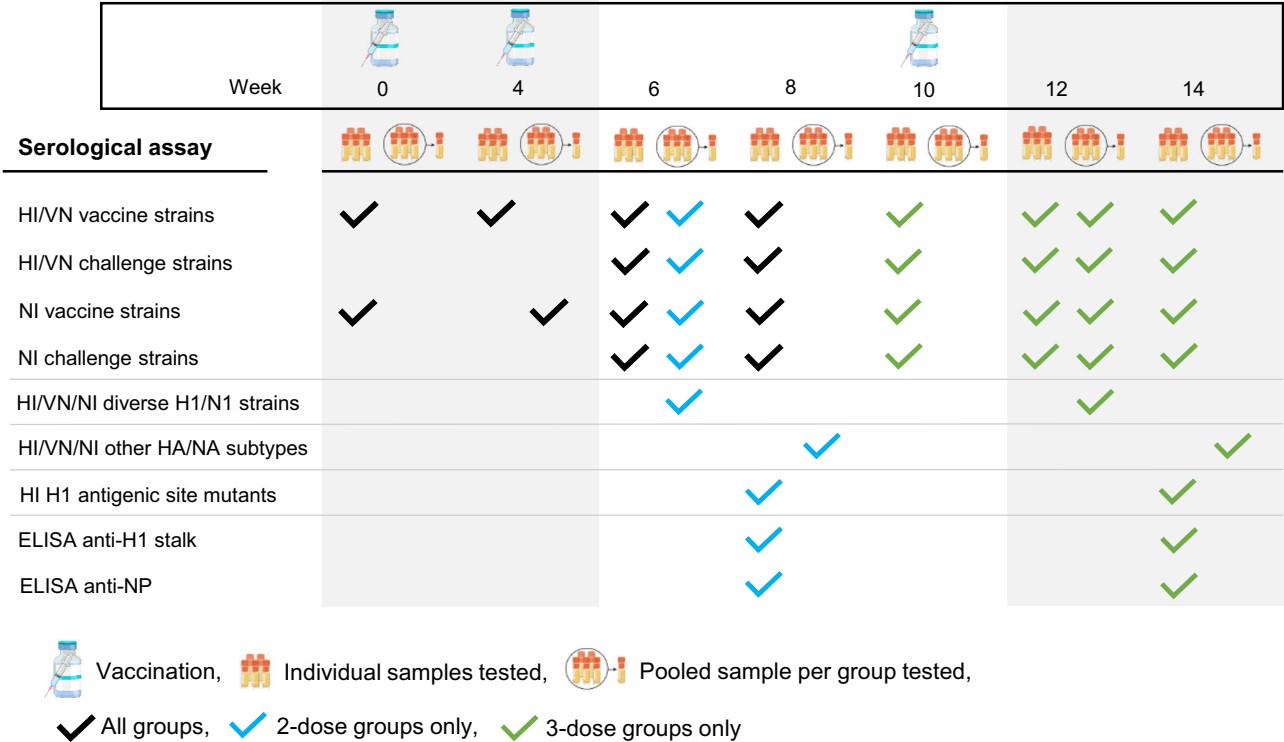

**Fig. 8 | Overview of serological assays performed with time points and use of individual and/or pooled serum samples.** Serum samples of individual pigs were used in most assays. A pooled serum sample of each of the 20 groups was used for functional assays (HI, VN, NI) against the panels of antigenically diverse H1/N1 strains (Fig. 3) and the panel of heterosubtypic strains (Fig. 5).

approximately half of the pigs were female and male respectively. The pigs were housed in a biosafety level 2 (BSL2) isolation facility with HEPA-filtered air. In experiment 1, eight two-dose heterologous prime-boost groups ($n = 5$) were primed with one of the four vaccine strains and boosted with a different strain 4 weeks later. Four homologous prime-boost groups ($n = 5$ or 10) received two administrations of the same vaccine strain. The vaccines were administered by deep intra-muscular injection into the neck. A mock-vaccinated challenge control group ($n = 10$) received two administrations of PBS with adjuvant. Four additional pigs served as unvaccinated unchallenged controls to assess histological characteristics of the respiratory tract of healthy pigs of the same cohort.

In experiment 2, we explored heterologous prime-boost vaccination regimens with three vaccine strains: CA09, ARM08 and G10. A two-dose heterologous prime-boost group (experiment 2a, $n = 7$) was primed with G10 and boosted 4 weeks later with bivalent ARM08 + G10 vaccine. A three-dose heterologous prime-boost group (experiment 2b, $n = 7$) was vaccinated sequentially with G10, ARM08 and CA09 at 4- and 6-week intervals. Three homologous prime-boost groups ($n = 7$) received two or three administrations of matched trivalent vaccine (TIV) or three administrations of monovalent ARM08 vaccine. Two mock-vaccinated control groups ($n = 6$) were injected two or three times with PBS and adjuvant.

Serum samples were collected on the day of each vaccination and 14 and 28 days after the booster vaccination(s). All sera were examined in HI, VN and NI assays against the four vaccine strains and the final, pre-challenge sera were also examined against both swine IAV strains used for challenge. Sera collected 14 days after the last vaccination, when we expect antibody titers to peak, were pooled per group and tested for functional antibodies against panels of antigenically diverse H1 and N1 strains. Sera collected 28 days after the final vaccination, before challenge, were pooled and tested in HI, VN and NI assays against heterosubtypic IAV strains. Individual pre-challenge sera were

also examined for antibodies against the HA stalk and NP, and for HI antibody targets in the HA head (Fig. 8).

In experiment 2, we also performed ELISpot assays to determine the frequency of IgG ASC specific for the three vaccine strains used and both challenge strains. For this purpose, we collected heparinized whole blood for isolation of PBMC at the time of the booster vaccination(s) and 7 days later.

One month after the last vaccination, 70 pigs of experiment 1 and all pigs of experiment 2 were challenged intranasally with 7.0 log$_{10}$ TCID$_{50}$ of the heterologous H1N1 swine IAVs A/swine/Ohio/511445/2007 (OH07) or A/swine/Italy/7704/2001 (IT01) (Fig. 2). Each vaccination-challenge group was housed in a separate BSL2+ isolation room. Three days post challenge, all pigs were euthanized to evaluate virus titers in nasal swabs, and 20% tissue homogenates of the nasal mucosa, trachea and samples of the right and left lung, as well as (histo)pathological lesions of the trachea and lungs.

### Ethics statement
All animal experiments were performed at the Faculty of Veterinary Medicine, Ghent University, and approved by its Ethical Committee (project identification codes: EC2015-119 and EC2018-70). Experiments comply with the European Union legislation on the protection of animals used for scientific purposes (directive 2010/63/EU).

### Vaccines and viruses
We prepared whole inactivated H1N1 IAV vaccines based on four antigenically distinct virus strains of the three known swine H1 lineages. The classical swine lineage strain A/California/04/2009 (CA09, clade 1A.3.3.2) is the prototype H1N1pdm09 virus and descendants of this virus are circulating in both humans and swine. A/swine/Cotes d'Armor/0046/2008 (ARM08, clade 1B.1.2.3) and A/swine/Illinois/00685/2005 (IL05, clade 1B.2.1) are human seasonal swine lineage strains from Europe and North America. A/swine/Gent/28/2010 (G10,

clade 1C.2.1) is a Eurasian avian lineage strain. Vaccine virus stocks were propagated in Madin-Darby canine kidney (MDCK, ATCC, catalog #CCL-34) cells and inactivated with ultraviolet light (UV) as previously described[28]. For this purpose, 10 ml aliquots of virus were thawed, placed in 60 mm Petri dishes to a fluid depth of 5 mm and exposed to 120 J/cm² from a UV source (Ultra-Violet Products Ltd., Cambridge, UK) for 10 min. The dishes were placed on ice at 10 cm distance from the light source. Loss of viral infectivity was confirmed by two serial passages in MDCK cells. Each 2 ml vaccine dose contained 256 hemagglutinating units (HAU) of a single vaccine strain, or of each of two or three different strains, diluted in PBS and 20% commercial oil-in-water adjuvant (Emulsigen®, MVP Laboratories, NE, USA). Thus, the total antigenic mass in the bivalent and trivalent formulations was 2- and 3-fold higher than in the monovalent vaccines.

The challenge virus strains A/swine/Ohio/511445/2007 (OH07, clade 1A.3.3.3) and A/swine/Italy/7704/2001 (IT07, clade 1B.1.2.2) represent lineage 1A and 1B swine IAVs from other clades than the vaccine strains (Fig. 1, Supplementary Table 1).

The antigenically diverse H1 and N1 virus strains for use in HI/VN and NI assays are shown in Fig. 3. The 24-member HI/VN panel (Fig. 3a, b) contained 19 H1N1 or H1N2 swine IAVs, including the vaccine and challenge strains, and five human seasonal H1N1 virus strains from 1934 to 2007. The swine IAV strains involved five of the seven clades that accounted for 87% of the H1 viruses in swine populations worldwide between 2010 and 2016, and eight clades with ≥10.0% average pairwise nucleotide distance (APD) from these dominant clades[18]. For clades with ≥9.5% within-clade APD, we selected at least two virus strains. The 14-member NI panel (Fig. 3c) comprised 12 H1N1 IAV strains that were part of the HI/VN panel, as well as one swine H3N1 and one avian H5N1 virus strain. It was representative of all four N1 lineages: avian, classical swine, human and pdm09.

To determine the genetic relationship between the surface proteins from the vaccine strains and the strains used for challenge and serology, HA1 and NA nucleotide sequences were downloaded from GenBank and translated to amino acids (aa) using MEGA version 7.0[54]. Amino acid differences between pairs of strains were determined after alignment with ClustalW, using MEGA 7.0 and R version 3.2.2[55]. The Jones-Taylor-Thornton model and nearest-neighbor interchange method were used to construct maximum likelihood trees. To determine the antigenic relationship, we performed cross-HI and cross-NI assays using post-vaccination pig sera against swine IAVs and post-infection ferret sera against human seasonal viruses. Three antigenic distance measures were calculated for both surface proteins of each pair of strains: (1) The P sequence value[56] is the fraction of different aa in the HA1 domain (326 or 327 aa) of the HA or in the NA (469 or 470 aa); (2) The P all antigenic site value[56] is the fraction of different aa in all putative antigenic sites of the HA1 (50 aa[3,57]) or the NA (193 aa[58]); (3). The antigenic distance is measured in antigenic units as described by Peeters et al.[59] One antigenic unit represents a 2-fold difference in HI or NI titer.

The heterosubtypic IAVs that were used in HI, VN and NI assays included two contemporary H3N2 swine IAVs (A/swine/Gent/172/2008, A/swine/PA/A01076777/2010) and six low pathogenic avian influenza viruses of subtype H2N3 (A/mallard/Alberta/205/1998), H5N1 (A/duck/Minnesota/1525/1981, A/mallard/Italy/3401/2005), H5N2 (A/chicken/Belgium/150/1999), H5N9 (A/chicken/Italy/22A/1998), and H7N1 (A/chicken/Italy/1067/1999).

Virus strains were obtained through the US Department of Agriculture (USDA) swine influenza repository held at the National Veterinary Service Laboratories (USA), the Worldwide Influenza Centre at the Francis Crick Institute (London, UK), the Centers for Disease Control (Atlanta, USA), the Ploufragan-Plouzané-Niort Laboratory of the French Agency for Food, Environmental and Occupational Health & Safety (ANSES, Ploufragan, France), Sciensano (Brussels, Belgium) and the Istituto Zooprofilattico Sperimentale delle Venezie (IZSLER, Legnaro, Padua, Italy).

## HI, VN and NI assays

All sera were heat inactivated (56 °C, 30 min) before use. The HI assay was performed according to standard procedures[60]. To remove non-specific inhibitors of the agglutination, the sera were treated with receptor-destroying enzyme (RDE) from *Vibrio cholerae* (Sigma-Aldrich, St. Louis, MO, USA, catalog #c8772) at a 4:1 ratio (serum:RDE) overnight at 37 °C. The next day, the remaining RDE was inactivated by the addition of 1.5% sodium citrate buffer at a 1:3 ratio (serum:buffer) and 30 min heating in a 56 °C water bath. The RDE-pretreated sera were then adsorbed onto 50% turkey erythrocytes at a 10:1 ratio (serum:erythrocytes) to remove natural serum agglutinins. After 1 h incubation at 4 °C, the virus-erythrocyte mixtures were centrifuged (1000 × g). Serial 2-fold dilutions of these pretreated sera were prepared in a U-bottom microtiter plate and an equal volume of 4 HAU of virus was added to each well. After incubation at room temperature for 1 h, 0.5% turkey erythrocytes were added to each well and hemagglutination patterns were read after 1 h.

In the VN assay, sera were 2-fold serially diluted in duplicate wells of a microtiter plate and 100 TCID₅₀ of virus was added in each well[61]. After 1 h incubation at 37 °C, MDCK cells were added to the virus-serum mixture at a concentration of 800,000 cells per ml. After 24 h incubation at 37 °C, the cells were air dried and fixed with 4% paraformaldehyde. Virus-positive MDCK cells were demonstrated by immunoperoxidase staining, using monoclonal antibodies against influenza virus nucleoprotein (HB-65 supernatant, 1:50 dilution, ATCC, catalog #H16-L10-4R), goat anti-mouse immunoglobulins conjugated with horseradish peroxidase (1:1000 dilution Agilent Technologies, Santa Clara, CA, USA, catalog #P044701-2), and 3-Amino-9-Ethylcarbazole (AEC, Sigma-Aldrich, catalog #A5754).

NI antibodies were measured by an enzyme-linked assay (ELLA) as described by Couzens et al.[62] In assays with pooled sera, the samples were pretreated with receptor destroying enzyme from *Vibrio cholerae* (Sigma-Aldrich, catalog #c8772) to eliminate non-specific inhibitors[63]. Briefly, serial 2-fold dilutions of sera were added to duplicate wells of a microtiter plate coated with fetuin (Sigma-Aldrich, catalog #F3385). A predetermined amount of the test strain was then added, and the mixture was incubated overnight at 37 °C. Each plate also had eight wells containing virus only (positive control) and eight wells containing PBS only (background control). After six washes with PBS-0.05% Tween 20 (PBS-T), peroxidase-conjugated peanut agglutinin (PNA, Sigma-Aldrich, catalog #L7759) was added to detect galactose exposed by NA-induced removal of the sialic acids on fetuin. The plates were incubated at room temperature for 2 h in the dark. The plates were then washed three times with PBS-T and O-phenylenediamine dihydrochloride (OPD) substrate (Sigma-Aldrich, P8287) was added for color development. The color reaction was stopped after 10 min by the addition of 1 N sulfuric acid. The amount of bound PNA correlates directly with the amount of sialic acid removed from the substrate, and thus with NA activity, and was quantified by optical density (OD) measurements at 490 nm using a Multiskan FC microplate reader (Thermo LabSystems). The % NA inhibition of the test wells was calculated as described by Couzens et al.[62]

Two recombinant viruses containing a mismatched HA (kindly provided by Dr. Xavier Saelens, VIB-UGent Center for Medical Biotechnology, and Dr. R. Webster, St. Jude Children's Research Hospital, Memphis, TN) were tested in parallel with the matched wild type PR34 and CA09 strains to check for interference of HA-specific antibodies. These viruses were 6:2 reassortants consisting of six gene segments from A/Puerto Rico/8/1934 (PR8 or PR34, H1N1) in conjunction with an H6 HA gene derived from A/mallard/Sweden/81/2002, whereas the NA gene was derived from PR8 (H6-PR8) or from A/Belgium/145-MA/2009 (H1N1) (H6-Bel09). The NA coding sequence of A/Belgium/145-MA/2009 (H1N1) has been deposited into Genbank (accession number: KJ867564.1) and is identical to the NA coding sequence of CA09, except for a 12 aa residue deletion in the stalk region of NA. The

average fold difference between NI titers against reassortant and wild type (wt) viruses was <4-fold for CA09 ($p = 0.18$, two-sided Mann–Whitney $U$ test) and <8-fold for PR34 ($p = 0.0019$).

Starting dilutions were 1:10 in the HI and NI assay, and 1:4 in the VN assay. Antibody titers were expressed as the reciprocal of the highest serum dilution that inhibited hemagglutination or virus replication, or that gave 50% inhibition of NA activity.

### ELISpot analysis of antibody-secreting cells (ASC)

An ELISpot assay to determine virus-specific IgG ASC was performed as described by Kitikoon et al.[64]. Briefly, 96-well ELISpot filter plates (Merck Millipore, Molsheim, France, catalog #MAIPS4510) were coated overnight at 4 °C with 200 HAU per well of live purified IAV. The next day, PBMC were isolated from whole blood samples by Ficoll-paque (GE health care, Uppsala, Sweden) density gradient centrifugation, resuspended in complete RPMI medium and dispensed ($5 \times 10^5$ cells per well) into duplicate wells of the plates. After 18 h incubation (37 °C, 5% $CO_2$), biotinylated mouse anti-porcine IgG monoclonal antibody (MT424, 1:1000 dilution, MABTECH Inc., Cincinnati, OH, USA, catalog #3151-6-1000) was added for 2 h to capture the secreted antibodies. Following 1 h incubation with streptavidin horseradish peroxidase (MABTECH, catalog #3310-9-1000), IgG ASC were visualized with 3,3′,5,5′-tetramethylbenzidine (TMB) substrate for ELISpot (MABTECH, catalog #3651-10). Blue spots, corresponding to activated ASC, were counted. Non-specific spots detected in wells coated with mock-infected, purified MDCK cell medium were subtracted from the counts of influenza-specific ASC.

### HI assays with H1 antigenic site mutants

We also performed HI assays against a panel of viruses with mutations in known HI sensitive epitopes in the H1 head domain. Five mutant viruses (H1-ΔSa, H1-ΔSb, H1-ΔCa1, H1-ΔCa2, H1-ΔCb) had an HA encoded by A/Michigan/45/2015 (MI15) in which one of the five classically defined H1 antigenic sites (Sa, Sb, Ca1, Ca2, Cb) were substituted with heterologous antigenic sites from either H5 or H13 HAs. MI15 was the human H1N1 influenza vaccine strain from 2017 to 2019 and has four aa substitutions in antigenic sites when compared to vaccine strain CA09. An additional mosaic H5/1 virus (mH5/1), where all five classically defined H1 antigenic sites were replaced with H5 antigenic sites, was used to detect HI antibodies against non-traditional epitopes in the HA head. The mutant virus panel has been described in detail by Liu et al.[6] HI assays against the six mutant viruses and wt MI15 were performed with chicken erythrocytes. The HI dominance index represents the fold reduction in HI titer against a specific mutant virus as compared to the titer against wt MI15 and is proportionate with the contribution of that antigenic site to the overall HI response.

### ELISA for antibodies against the H1 stalk and nucleoprotein

We measured anti-H1 stalk antibodies using a recombinant chimeric cH6/1 HA antigen with the H6 head domain from A/mallard/Sweden/81/2002 (H6N1) and H1 stalk domain from CA09[65]. Recombinant NP derived from A/Michigan/45/2015 (MI15) was used to detect antibodies against the NP[66]. We used a classical ELISA in which 96-well clear-bottom plates (Immulon 4HBX, Thermo Scientific) were coated overnight at 4 °C with either recombinant protein at a concentration of 2 μg/ml. The next day, plates were washed three times with PBS containing 0.1% Tween 20 (PBS-T) and blocked with 3% milk diluted in PBS-T for 1 h at room temperature. Blocking buffer was then removed and serum samples diluted 1:100 in PBS-T containing 1% milk were added to the plates and serially diluted 2-fold to a final dilution of 1:12800. After 2 h incubation and three washes with PBS-T, a secondary peroxidase-labeled rabbit anti-pig IgG antibody (1:3000 dilution, Sigma-Aldrich, catalog #A5670) was added to detect serum antibodies attached to the antigen. Plates were left at room temperature for 1 h and then washed

four times with PBS-T with a shaking step included. O-phenylenediamine dihydrochloride (OPD) substrate (SigmaFast OPD, Sigma-Aldrich, catalog #P9187) developing solution was added to the plates for 10 min. The reaction was stopped with 3 M hydrochloric acid and the OD was measured at 490 nm with a Synergy 4 plate reader (BioTek). An endpoint titer was defined as the reciprocal of the highest serum dilution at which the OD remained greater than three standard deviations above the average of blank wells.

### Virus titration and lesion scores

Virus titrations were performed in MDCK cells[61]. Nasal swab transport medium and 20% tissue homogenates were first clarified by centrifugation. Ten-fold serial dilutions of samples were inoculated into four cell culture replicates in microtiter plates. MDCK cells were observed for development of cytopathic effect over 7 days. Virus titers were expressed as $\log_{10}$ $TCID_{50}$ per g tissue, or per 100 mg nasal secrete.

At necropsy, lungs were evaluated for the percentage of macroscopic pneumonia by assessing purple-red consolidation typical of swine IAV infection. The percentage of the surface affected with pneumonia was estimated visually for each lung lobe, and the total percentage for the entire lung was calculated based on the weighted proportion of each lobe relative to the total lung volume[67]. Samples from the trachea and right cardiac or affected lung lobe were fixed in 4% buffered formalin and processed for routine histopathological examination. Microscopic lesions of the lung and trachea were evaluated by a veterinary pathologist blinded to treatment groups and scored as described by Vincent et al.[68] (see Supplementary Table 9). Microscopic lung lesion scores were based on the severity of three parameters: (1) epithelial damage in intrapulmonary airways, (2) peribronchiolar lymphocytic cuffing, (3) neutrophil exudation in bronchioles and alveoli. Microscopic tracheal lesion scores were based on the severity of epithelial damage.

### Statistical analysis

Antibody titers against the vaccine strains, numbers of ASC and lesion scores were compared between groups using the Kruskal–Wallis $H$-test. Analysis of variance (ANOVA) was used to compare virus titers. Samples that tested negative in serological assays were assigned a value corresponding to half of the minimum detectable antibody titer. Samples that tested negative for virus were given a value of 0.85 $\log_{10}$ $TCID_{50}$/g. Dunn's (Kruskal-Wallis) and Dunnet's (ANOVA) tests were used for multiple comparisons. Spearman's rank correlation coefficient was used to assess the relationship between (a) antibody titers in different assays, (b) antibody titers before challenge and virus titers post challenge, (c) virus titers and lesion scores. Data were plotted using GraphPad Prism 9.5.0.

## Data availability

The datasets generated during and/or analyzed during this study are provided in the Supplementary Information and in the Source Data file. All data used in figures and tables are provided in these files. The hemagglutinin (HA) and neuraminidase (NA) sequences of the influenza A virus strains used are available in GenBank (see Supplementary Table 1 for accession numbers). Source data are provided with this paper.

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

## Acknowledgements

The authors thank Melanie Bauwens and Nele Dennequin for laboratory technical assistance and Zeger Van den Abeele and Jonathan Vandenbogaerde for assistance with animal studies. We thank several colleagues and organisations for sharing IAV strains: John McCauley (Francis Crick Institute), Emanuela Foni (ISZLER), Xavier Saelens (VIB-UGent), Gaëlle Simon (ANSES), Sabrina Swenson (USDA), Thierry van den Berg (Sciensano), Amy Vincent (USDA), Robert Webster, the USDA swine influenza repository, and the pandemic influenza preparedness framework (WHO). Special thanks go to Ruth Mwende Mumo for help with figures. Funding was provided by the Belgian Federal Public Service for Health, Food Chain Safety and Environment (RF 16/6305, EVAFLU), and by the University of Ghent Special Research Fund (grant no. 01J102017). Partial support was provided by NIH 5 P01 AI097092 (PP), by NIH 1 R01 AI145870 (PP), by NIAID SINAI-EMORY CIVIC/RESEARCH - Collaborative Influenza Vaccine Innovation Centers (CIVICs) 75N93019C00051/OPTION 2 (PP), by an NIAID Centers of Excellence for Influenza Research and Response (CEIRR) 75N93021C00014 (PP) and by DoD grant GMP Production of Candidate Pan-Group 2 Influenza A Virus W81XWH1810488 (PP). Published with the support of the University Foundation of Belgium.

## Author contributions

K.V.R., A.P. and F.K. designed research; A.P., J.C.M.G., I.T. and E.V. performed the animal experiments; K.C. performed histopathological analyses; K.V.R., A.P., J.C.M.G., I.T., P.M., F.K. and E.V. analysed and interpreted the data; P.M., S.L., P.P. and F.K. contributed novel reagents and tools; K.V.R., F.K. and E.V. wrote the manuscript with input from P.P.

## Competing interests

The Icahn School of Medicine at Mount Sinai has filed patent applications for various influenza virus vaccine candidates which list F.K. and P.P. as inventors. All other authors declare no competing interests.
