## [Peer Review File · Nature Communications]

REVIEWER COMMENTS

Reviewer #1 (Remarks to the Author):

In the manuscript by Van Reeth et al. the authors performed a comprehensive evaluation of homologous and heterologous prime-boost strategies of whole virus inactivated vaccines for swine influenza. The results show that broadly reactive and cross-protective antibodies against diverse IAV-S H1 viruses are obtained only after 3 immunizations with monovalent WIV vaccine candidates. Although this is an important observation, it does not represent a completely novel finding in the influenza field as there are other studies demonstrating that heterologous prime boost regimens elicit broader cross-reactivity and protection against influenza viruses. Because of the complexity of the experimental design, the manuscript and figures are difficult to follow. Below I provide some specific comments that may help the authors to improve the manuscript:

- 1. Antibody titers should be presented in titers and not scores. The scoring system presented is confusing and the use of titer and score interchangeably in the text complicates interpretation of the results.**
- 2. All antibody scores were determined with pooled serum samples. This may mask experimental variability which is often observed in immunization studies in swine. Each animal in the treatment groups should be tested, antibody titers presented and group geometric mean titers presented in the figures.**
- 3. the graphical representation of the data needs to be improved. The table format with the scores and color scheme is very difficult to follow. Bar graphs showing individual animal Ab titers would likely be a better way of presenting the data.**

Reviewer #2 (Remarks to the Author):

Van Reeth et al is an interesting study with noteworthy results in which responses to different homologous and heterologous vaccine regimens are assessed. Of particular interest is the three-dose heterologous regimen which produces a broadly cross-reactive H1 subtype response mediated by HA head domain epitopes. This response is also protective. The study is rigorous with sound methodology; antibody and APC responses are assessed using multiple assays, in addition to challenge studies.

The study is significant to the field because (i) it demonstrates that epitopes in the HA head domain can induce broadly cross-reactive responses via vaccination, and (ii) presents an intriguing approach to vaccination, which could improve protection against influenza in swine and humans.

In addition to this, as the authors elude, the study raises many further questions, such as which HA head epitopes outside of the antigenic sites and stem mediate the broadly reactive response induced by the three dose vaccination regimen.

Overall the study is thorough and well-written. I have several minor comments, which I have outlined below.

Minor comments:

1. The title is quite hard to read in its current form, and in my opinion, doesn't do the paper justice. Perhaps 'Generation of multi-clade...', or the inclusion of "produced/induced by" instead of just "by"?

2. Line 59, page 3: suggest reconsidering the use of "on the other hand".

3. Line 94, page 5 "determine" to 'assess' or 'analyse'.

4. Line 145, page 7, use of "per se concordant" is confusing. I would suggest re-phrasing.

5. The authors present an alternative vaccination strategy, which their data supports. However, prior immunity is likely to influence responses to conserved or semi-conserved head epitopes targeted by the three-dose heterologous vaccine regimen. Although the authors repeatedly mention the 2009 pandemic as an example of prior immunity to influenza being beneficial, it might be useful to address the suitability of the three-dose regimen to human vaccination in relation to prior immunity in the discussion. For example, some studies suggest prior immunity could be a hindrance via an original antigenic sin (OAS) mechanism (eg PNAS 2017 Nov 21;114(47):12578-12583). This could impact the effectiveness of the proposed three-dose heterologous vaccination regimen.

Overall, an excellent and intriguing study.

Craig Thompson, Warwick Medical School, University of Warwick, UK

Reviewer #3 (Remarks to the Author):

This is an excellent manuscript by Van Reeth et al. By using a highly relevant and biologically significant animal model of influenza the manuscript offers additional insights into how vaccination regimes affect the breadth of immune responses against influenza. Through a series of studies using 2 dose or 3 dose vaccination regimes using UV-inactivated/adjuvanted vaccine candidates against H1 subtype swine influenza strains. The data is consistent with what the investigators statement that "Three doses of heterologous monovalent H1N1 vaccine result in seroprotective neutralizing antibodies against 71% of a diverse panel of human and swine H1 strains, detectable antibodies 31 against 88% of strains, and sterile cross-clade immunity against two heterologous challenge strains." However, it is less clear that (the three dose heterologous strategy) "...is as good or better than giving three doses of matched trivalent vaccine". The reason for the latter statement to be less convincing comes from the lack of details on the amount of antigen in the TIV formulation; if it is prepared as it is assumed, then the amount of each antigen in the TIV is $\frac{1}{3}$ of the amount delivered for each antigen with each monovalent vaccine. Given the relatively minor differences between serological responses from pigs in the 3XTIV group versus the G10-ARM08-CA09 group, but higher ASC levels in the latter; is it possible that such differences are due to lower amount of each antigen/dose in the TIV formulation? I believe the authors need to address this confusion to strengthen the conclusions in the manuscript. Fig 5 and 6 are the core of the ms and highly significant to the ms. However, there aspects of those figures that perhaps the authors would like to discuss further: 1) the 3XTIV group show less variations among individuals than the G10-ARM08-CA09 group, making the former a more reliable approach to vaccination against H1 viruses, 2) Although not considered to be neutralizing, anti-NP antibodies have been shown to confer some protection against influenza, anti-NP ELISA titers could add to the

overall discussion and to establish whether those can be linked to protection after challenge.

Minor comments

Page 4, In 69-71: "Thus, people born after 1977 initially had little or no cross-reactive antibodies, but they rapidly developed a broad, pan-H1 neutralizing antibody response upon infection or vaccination with H1N1pdm09, leading to the extinction of earlier seasonal H1N1 viruses [8,12-14]" Can you really say that development of a panH1 neutralizing response led to the extinction of the earlier seasonal H1N1. Interesting hypothesis, but there is no hard evidence. You might want to consider revising this statement.

Page 6, In 130-131: "After the first vaccination, antibodies were either undetectable or detectable against..." How soon were these evaluated? 14 days after first vaccination?

Fig 1 legend: "P sequence values are shown in the upper right triangle, antigenic units in the lower left triangle" This is not obvious to everyone. Please consider adding notations on the figure itself

Response to reviewers:

Dear reviewers,

We want to thank you for your constructive criticism and for giving us the opportunity to submit a revised manuscript. Below are the responses to each of your comments, in blue. We feel that the manuscript has been substantially improved in the revision process. Many thanks for your time and effort.

Sincerely,

Kristien Van Reeth

Reviewer #1

In the manuscript by Van Reeth et al. the authors performed a comprehensive evaluation of homologous and heterologous prime-boost strategies of whole virus inactivated vaccines for swine influenza. The results show that broadly reactive and cross-protective antibodies against diverse IAV-S H1 viruses are obtained only after 3 immunizations with monovalent WIV vaccine candidates. Although this is an important observation, it does not represent a completely novel finding in the influenza field as there are other studies demonstrating that heterologous prime boost regimens elicit broader cross-reactivity and protection against influenza viruses. Because of the complexity of the experimental design, the manuscript and figures are difficult to follow. Below I provide some specific comments that may help the authors to improve the manuscript:

Thank you for these helpful comments. Please find the response to each of them below.

(1) Antibody titers should be presented in titers and not scores. The scoring system presented is confusing and the use of titer and score interchangeably in the text complicates interpretation of the results.

We have left out the scoring system throughout the revised manuscript and now only use antibody titers. The text, tables and figures have been revised accordingly, as explained under point 2 and 3.

(2) All antibody scores were determined with pooled serum samples. This may mask experimental variability which is often observed in immunization studies in swine. Each animal in the treatment groups should be tested, antibody titers presented and group geometric mean titers presented in the figures.

We agree that it is ideal to present antibody titers of individual animals. The use of pooled sera, however, may be a solution in case of large numbers of serum samples and/or multiple serological assays and test strains. This practice has been used before in vaccination studies with influenza and HIV, and in clinical studies with SARS-CoV2. In the present study, we had serum samples from 121 individual pigs at 4 or 7 different timepoints. We used 5 different serological assays. The 3 functional assays (HI, VN and NI), which are very labor intensive, were also performed against large panels of heterologous H1/N1 strains and of heterosubtypic strains. Only for this set of assays we used pooled sera per group. We apologize if this was unclear in the original manuscript.

Functional assays against the most important strains and antigens were performed with individual sera. These include assays against the 4 vaccine and both challenge strains, 5 mutant H1 strains and a mosaic H5/1 strain. The table below (Table 4 in the manuscript) summarizes the assays performed at each timepoint and whether individual or pooled sera were used.

We have now also performed additional HI, VN and NI assays to compare antibody titers obtained with individual and pooled sera for the 6 most important strains at 14 days after the final vaccination. The differences between antibody titers of pooled sera and geometric mean titers of individual sera are shown in the file “Addendum for reviewers - Comparison Ab titers pooled versus individual sera”. While pooled sera generally had higher titers, the differences were ≤ 2 -fold for most samples (81% of samples in HI assay, 86% in VN, and 80% in NI). The comparative table could be included in the Supplementary tables if the reviewer wishes so.

Table 4. Overview of serological assays performed with time points and use of individual and/or pooled serum samples

Serological assay	Time points in days post vaccination (dpv) 1, 2 or 3												
	0	28 dpv1		14		28		42 dpv2		14		28 dpv3	
HI/VN vaccine strains	✓	✓	✓	✓	✓	✓	✓	✓	✓	✓	✓	✓	✓
HI/VN challenge strains			✓	✓	✓	✓	✓	✓	✓	✓	✓	✓	✓
NI vaccine strains	✓		✓	✓	✓	✓	✓	✓	✓	✓	✓	✓	✓
NI challenge strains			✓	✓	✓	✓	✓	✓	✓	✓	✓	✓	✓
HI/VN/NI diverse H1/N1 strains				✓						✓			
HI/VN/NI other HA/NA subtypes						✓							✓
HI H1 antigenic site mutants					✓							✓	
ELISA anti-H1 stalk					✓							✓	
ELISA anti-NP					✓							✓	

 Individual samples tested,
  Pooled sample per group.
 ✓ Results available for all groups,
 ✓ 2-dose groups only,
 ✓ 3-dose groups only

(3) the graphical representation of the data needs to be improved. The table format with the scores and color scheme is very difficult to follow. Bar graphs showing individual animal Ab titers would likely be a better way of presenting the data.

The table below presents an overview of all data, how they were presented in the original manuscript and how we tried to improve this in the revised manuscript. Because of the multitude of individual antibody titers, we did not manage to present all of them in bar graphs. The most important results, however, are presented as bar graphs showing individual data. These include 1) numbers of ASC and VN and NI titers against the

vaccine and challenge strains in experiment 2 (Fig. 3), 2) HI titers against H1 head mutants (Fig. 5), 3) virus titers post challenge (Fig. 6).

Data	Original manuscript	Revised manuscript
HI/VN/NI Ab titers against vaccine (n=4) and challenge strains (n=2); 28 d after final vaccination – individual sera	Fig. 2: Geometric mean Ab titers per group converted to scores (0-5)	Table 2: Geometric mean Ab titers per group as numeric values Individual Ab titers, as well as geometric mean Ab titers \pm SD are available in the Excel file with “Source data”
HI/VN/NI Ab titers against panel of diverse H1 (n=24) and N1 (n=14) strains; 14 d after final vaccination – pooled sera	Fig. 3: Antibody titers of pooled sera converted to scores and color coded	Fig. 2: Antibody titers of pooled sera converted to color gradient heatmaps Ab titers shown as numeric values in Source data Interpretation of Ab titers in Supplementary Table 8
Ab secreting cells (ASC) post vaccination 2 and/or 3 (expt 2) and association with VN/NI Abs - individual samples	Fig. 4: Mean numbers of ASC per group shown as bars \pm SD Geometric mean VN and NI Ab titers converted to scores and color coded	Fig. 3: Mean numbers of ASC per group shown as bars, individual numbers as dots Geometric mean VN and NI titers shown as bars, individual titers as dots All data, including SD, available in Source data
HI/VN/NI Ab titers against other HA/NA subtypes (n=8); 28 d after final vaccination – pooled sera	Table 2: Antibody titers of pooled sera converted to scores and color coded	Fig. 4: Antibody titers of pooled sera converted to color gradient heatmaps Ab titers shown as numeric values in Source data
ELISA titers against H1 stalk; 28 d after final vaccination – individual sera	Table 2: Geometric mean Ab titers per group \pm SD as numeric values	Fig. 4: Geometric mean Ab titers shown as bars, individual titers as dots Individual Ab titers, geometric means \pm SD available in Source data
ELISA titers against NP; 28 d after final vaccination – individual sera	Not included	Fig. 4: Geometric mean Ab titers shown as bars, individual titers as dots Individual Ab titers, geometric means \pm SD available in Source data
HI Ab titers against H1 strains with mutations in HA head (n=5) and mosaic H5/1 virus; 28 d after final vaccination – individual sera	Fig. 5: Geometric mean Ab titers per group shown as bars, individual titers as dots	Fig. 5: Improved lay-out with bars in color Individual Ab titers, geometric means \pm SD available in Source data Table has been removed

		from the figure and inserted in manuscript as Table 3
Virus titers post challenge	Fig. 6: Geometric mean virus titers per group \pm SD, numbers of virus-positive pigs	Fig. 6: Geometric mean virus titers shown as bars, individual titers as dots Individual virus titers, geometric means \pm SD available in Source data
Supplementary Tables	Table 8 and 9: antibody titers expressed as scores	New Supplementary Tables 8 and 9

Please note that for the sake of clarity, we did not add error bars to the bar graphs, but standard deviations can be found in the Source Data files.

We have made a few other changes to clarify the experimental design and graphical representations for the reader:

- Table 4. "Overview of serological assays, timepoints and use of individual or pooled sera" has been added to the manuscript.
- Fig. 3 includes a schematic of the prime-boost regimens in experiment 2.
- The same structure and listing of experimental groups are used to represent serology results in Table 2, Fig. 2 and Fig. 4.

Reviewer #2

Van Reeth et al is an interesting study with noteworthy results in which responses to different homologous and heterologous vaccine regimens are assessed. Of particular interest is the three-dose heterologous regimen which produces a broadly cross-reactive H1 subtype response mediated by HA head domain epitopes. This response is also protective. The study is rigorous with sound methodology; antibody and APC responses are assessed using multiple assays, in addition to challenge studies.

The study is significant to the field because (i) it demonstrates that epitopes in the HA head domain can induce broadly cross-reactive responses via vaccination, and (ii) presents an intriguing approach to vaccination, which could improve protection against influenza in swine and humans.

In addition to this, as the authors elude, the study raises many further questions, such as which HA head epitopes outside of the antigenic sites and stem mediate the broadly reactive response induced by the three dose vaccination regimen.

Overall the study is thorough and well-written. I have several minor comments, which I have outlined below.

Minor comments:

(1) The title is quite hard to read in its current form, and in my opinion, doesn't do the paper justice. Perhaps 'Generation of multi-clade...', or the inclusion of "produced/induced by" instead of just "by"?

We agree with the reviewer. It was a challenge to find a suitable title of no more than 15 words. We have now changed the title into “Sequential vaccinations with divergent H1N1 influenza virus strains induce multi-H1 clade neutralizing antibodies in swine”

(2) Line 59, page 3: suggest reconsidering the use of “on the other hand”.

Has been replaced by “Unfortunately”

(3) Line 94, page 5 “determine” to ‘assess’ or ‘analyse’.

Has been replaced by “assess”

(4) Line 145, page 7, use of “per se concordant” is confusing. I would suggest re-phrasing.

This sentence has been rephrased as “Post-vaccination NI antibody titers followed similar kinetics as HI/VN titers, but they were higher and showed broader cross-reactivity.”

(5) The authors present an alternative vaccination strategy, which their data supports. However, prior immunity is likely to influence responses to conserved or semi-conserved head epitopes targeted by the three-dose heterologous vaccine regimen. Although the authors repeatedly mention the 2009 pandemic as an example of prior immunity to influenza being beneficial, it might be useful to address the suitability of the three-dose regimen to human vaccination in relation to prior immunity in the discussion. For example, some studies suggest prior immunity could be a hindrance via an original antigenic sin (OAS) mechanism (eg PNAS 2017 Nov 21;114(47):12578-12583). This could impact the effectiveness of the proposed three-dose heterologous vaccination regimen.

Thank you for this valuable suggestion. In the revised manuscript we mention the issue of prior immunity / original antigenic sin (OAS) in the final paragraph of the discussion. We did not go into detail because of the complexity of OAS and the many unknowns, but we refer to some highly relevant papers (refs. 47-50). One of these is an excellent review article (Linderman et al., 2021) in which the paper mentioned by the reviewer (Zost et al. PNAS 2017) is also discussed.

The following text was changed in / added to the discussion (Lines 421-436): “Our results cannot be directly extrapolated to humans, in which preexisting immunity will shape the antibody response to subsequent influenza virus exposures. The phenomenon of immune imprinting or “original antigenic sin” is complex and incompletely understood [47,48]. Depending on the context, it may interfere with or potentiate the response to subsequent influenza vaccinations [27,49,50]. Anyhow, the vastly different immune histories in people of different ages may impact the design and implementation of new prime-boost strategies in humans. Furthermore, influenza vaccines and vaccine policies also differ in swine versus humans. The strains for use in human influenza vaccine production are selected each year and generally show minimal antigenic differences with the previous year’s strain... Pigs are natural hosts for both influenza and coronaviruses and an excellent animal model to study the immune response to sequential vaccinations [20,52], as well as the effect of prior immunity [53].”

Overall, an excellent and intriguing study.

Reviewer #3

This is an excellent manuscript by Van Reeth et al. By using a highly relevant and biologically significant animal model of influenza the manuscript offers additional insights into how vaccination regimes affect the breadth of immune responses against influenza. Through a series of studies using 2 dose or 3 dose vaccination regimes using UV-inactivated/adjuvanted vaccine candidates against H1 subtype swine influenza strains. The data is consistent with what the investigators statement that "Three doses of heterologous monovalent H1N1 vaccine result in seroprotective neutralizing antibodies against 71% of a diverse panel of human and swine H1 strains, detectable antibodies 31 against 88% of strains, and sterile cross-clade immunity against two heterologous challenge strains."

- (1) However, it is less clear that (the three dose heterologous strategy) "...is as good or better than giving three doses of matched trivalent vaccine". The reason for the latter statement to be less convincing comes from the lack of details on the amount of antigen in the TIV formulation; if it is prepared as it is assumed, then the amount of each antigen in the TIV is $\frac{1}{3}$ of the amount delivered for each antigen with each monovalent vaccine. Given the relatively minor differences between serological responses from pigs in the 3XTIV group versus the G10-ARM08-CA09 group, but higher ASC levels in the latter; is it possible that such differences are due to lower amount of each antigen/dose in the TIV formulation? I believe the authors need to address this confusion to strengthen the conclusions in the manuscript.

Thank you for pointing out this confusion. The trivalent vaccine (TIV) contains the same amount of each antigen / vaccine strain (256 HAU) as the monovalent vaccines. We have now clarified this and added more details on the amount of antigen to the methods section.

See Lines 504-508 (Methods): "Each 2 ml vaccine dose contained 256 hemagglutinating units (HAU) of a single vaccine strain, or of each of two or three different strains, diluted in PBS and ... Thus, the total antigenic mass in the bivalent and trivalent formulations was 2- and 3-fold higher than in the monovalent vaccines."

Lines 326-328 (Results): "Three doses of heterologous monovalent vaccine were as effective as three doses of matched TIV vaccine and better than two doses, while the total amount of antigen per dose was 3-fold lower."

- (2) Fig 5 and 6 are the core of the ms and highly significant to the ms. However, there aspects of those figures that perhaps the authors would like to discuss further: 1) the 3XTIV group show less variations among individuals than the G10-ARM08-CA09 group, making the former a more reliable approach to vaccination against H1 viruses, 2) Although not considered to be neutralizing, anti-NP antibodies have been shown to confer some protection against influenza, anti-NP

ELISA titers could add to the overall discussion and to establish whether those can be linked to protection after challenge.

Indeed, for HI antibodies against HA head mutants and the mosaic virus, there is more variation in the G10-ARM08-CA09 group than in the 3xTIV group. For many other parameters, however, the variation in both groups is similar. These parameters include HI, VN and NI antibodies against vaccine and challenge strains, ELISA titers against the HA stalk and the NP, and virus titers. The change from means to individual values / dots in the revised Fig. 3, 4 and 6 should illustrate this point.

We thank the reviewer for the comment about antibodies to the nucleoprotein. The Krammer lab has now performed an anti-NP ELISA on all individual samples at 28 days after the final vaccination. The results are shown in the revised Fig. 4, next to the results of the anti-HA stalk ELISA. Anti-NP antibody titers were minimal (≤ 800) and the correlation with protection against challenge was weak. We have added just a few sentences about anti-NP antibody titers to the discussion, because of the uncertainties about their protective mechanisms and role, and conflicting data in the literature.

See also Lines 256-262 (Results): "We also tested individual pre-challenge sera by enzyme-linked immunosorbent assay (ELISA) for antibodies against the conserved HA stalk and nucleoprotein (NP). Mean IgG ELISA titers against both proteins were higher in three-dose than in two-dose vaccine groups (Fig. 4c, d), but there were no differences between homologous and heterologous prime-boost groups, and no associations with VN antibody titers. These results indicate that the neutralizing antibody response does not extend beyond the H1 subtype and mainly targets the head of the HA."

Lines 377-384 (Discussion): "Likewise, the NP is conserved between all IAV subtypes [38], but anti-NP antibody titers remained below those detected in adult humans or influenza-infected mice [39]. Anti-NP antibodies are not neutralizing and their role in protection remains controversial and may be poor [1,27,39]. While antibody titers against both proteins increased after a third vaccination of pigs, they were similar in homologous and heterologous prime-boost groups and only weakly correlated with protection against heterologous H1N1 challenge. All this supports that protection is limited to the H1 subtype and largely mediated by epitopes in the HA head domain."

Minor comments

- (3) Page 4, ln 69-71: "Thus, people born after 1977 initially had little or no cross-reactive antibodies, but they rapidly developed a broad, pan-H1 neutralizing antibody response upon infection or vaccination with H1N1pdm09, leading to the extinction of earlier seasonal H1N1 viruses [8,12-14]" Can you really say that development of a panH1 neutralizing response led to the extinction of the earlier seasonal H1N1. Interesting hypothesis, but there is no hard evidence. You might want to consider revising this statement.

This statement has been rephrased as "..., which has likely contributed to the extinction of earlier seasonal H1N1 viruses [8,12-14]"

- (4) Page 6, In 130-131: "After the first vaccination, antibodies were either undetectable or detectable against..." How soon were these evaluated? 14 days after first vaccination?

This has been amended: "One month after the first vaccination, antibodies were either undetectable or detectable against..."

- (5) Fig 1 legend: "P sequence values are shown in the upper right triangle, antigenic units in the lower left triangle" This is not obvious to everyone. Please consider adding notations on the figure itself

This has been done.

REVIEWERS' COMMENTS

Reviewer #2 (Remarks to the Author):

The authors have addressed all my comments.

Reviewer #3 (Remarks to the Author):

The authors have done an excellent job at responding the comments of previous reviewers and have significantly improved the manuscript. (I'd like to apologize to the authors for my late response. The e-mails regarding this revision assignment have been going to my spam folder and I did not see them until recently).